# The shared frameshift mutation landscape of microsatellite-unstable cancers suggests immunoediting during tumor evolution

Alexej Ballhausen [1,2,3,16], Moritz Jakob Przybilla [1,2,3,16], Michael Jendrusch[1,2,3,16], Saskia Haupt [4],
Elisabeth Pfaffendorf[1,2,3], Florian Seidler [1,2,3], Johannes Witt[1,2,3], Alejandro Hernandez Sanchez [1,2,3],
Katharina Urban[1,2,3], Markus Draxlbauer[1,2,3], Sonja Krausert[1,2,3], Aysel Ahadova[1,2,3], Martin Simon Kalteis [1,2,3],
Pauline L. Pfuderer [1,2,3], Daniel Heid[1,2,3], Damian Stichel[1,5], Johannes Gebert[1,2,3], Maria Bonsack [6,7,8],
Sarah Schott [9], Hendrik Bläker[10], Toni Seppälä [11], Jukka-Pekka Mecklin[12,13], Sanne Ten Broeke[14],
Maartje Nielsen[14], Vincent Heuveline [4], Julia Krzykalla[15], Axel Benner[15], Angelika Beate Riemer [6,7],
Magnus von Knebel Doeberitz [1,2,3] & Matthias Kloor [1,2,3✉]

The immune system can recognize and attack cancer cells, especially those with a high load of mutation-induced neoantigens. Such neoantigens are abundant in DNA mismatch repair (MMR)-deficient, microsatellite-unstable (MSI) cancers. MMR deficiency leads to insertion/deletion (indel) mutations at coding microsatellites (cMS) and to neoantigen-inducing translational frameshifts. Here, we develop a tool to quantify frameshift mutations in MSI colorectal and endometrial cancer. Our results show that frameshift mutation frequency is negatively correlated to the predicted immunogenicity of the resulting peptides, suggesting counterselection of cell clones with highly immunogenic frameshift peptides. This correlation is absent in tumors with *Beta-2-microglobulin* mutations, and HLA-A*02:01 status is related to cMS mutation patterns. Importantly, certain outlier mutations are common in MSI cancers despite being related to frameshift peptides with functionally confirmed immunogenicity, suggesting a possible driver role during MSI tumor evolution. Neoantigens resulting from shared mutations represent promising vaccine candidates for prevention of MSI cancers.

[1] Department of Applied Tumor Biology, Institute of Pathology, University of Heidelberg, Heidelberg, Germany. [2] Collaboration Unit Applied Tumor Biology, German Cancer Research Center (DKFZ), Heidelberg, Germany. [3] Molecular Medicine Partnership Unit (MMPU), Heidelberg University Hospital and EMBL, Heidelberg, Germany. [4] Engineering Mathematics and Computing Lab (EMCL), Interdisciplinary Center for Scientific Computing (IWR), Heidelberg University, Heidelberg, Germany. [5] Clinical Cooperation Unit Neuropathology, German Cancer Research Center (DKFZ), Heidelberg, Germany. [6] Immunotherapy and Immunoprevention, German Cancer Research Center (DKFZ), Heidelberg, Germany. [7] Molecular Vaccine Design, German Center for Infection Research (DZIF), partner site Heidelberg, Heidelberg, Germany. [8] Faculty of Biosciences, Heidelberg University, Heidelberg, Germany. [9] Department of Obstetrics and Gynecology, University Hospital Heidelberg, Heidelberg, Germany. [10] Institute of Pathology, University Hospital Leipzig, Leipzig, Germany. [11] Department of Gastrointestinal Surgery, Helsinki University Hospital and University of Helsinki, Helsinki, Finland. [12] Department of Education and Research, Central Finland Central Hospital, Jyväskylä, Finland. [13] Department of Sports and Health Sciences, University of Jyväskylä, Jyväskylä, Finland. [14] Department of Clinical Genetics, Leiden University Medical Center, Leiden, The Netherlands. [15] Division of Biostatistics, German Cancer Research Center (DKFZ), Heidelberg, Germany. [16] These authors contributed equally: Alexej Ballhausen, Moritz Jakob Przybilla, Michael Jendrusch. ✉email: matthias.kloor@med.uni-heidelberg.de

DNA mismatch repair (MMR) deficiency is a major mechanism causing genomic instability in human cancer. MMR-deficient cancers accumulate an exceptionally high number of somatic mutations. These mutations encompass certain types of single-nucleotide alterations but mostly insertion/deletion (indel) mutations at repetitive sequence stretches termed microsatellites (microsatellite instability (MSI))[1,2].

About 15% of colorectal cancers (CRCs), up to 30% of endometrial cancers (ECs) and multiple other tumors display the MSI phenotype[3]. MSI tumors can develop sporadically or in the context of Lynch syndrome, the most common inherited cancer predisposition syndrome. Due to this very specific process of genomic instability, the pathogenesis of MSI cancers can be precisely dissected[4]: indel mutations of coding microsatellites (cMSs), which almost exclusively affect coding mononucleotide repeats[5–7], in genomic regions encoding tumor-suppressor genes are considered major drivers of MSI tumorigenesis. Importantly, the same indels that inactivate tumor-suppressor genes simultaneously cause translational frameshifts, thereby generating unique frameshift peptides as a major source of neoantigens[4,8].

For the recognition of neoantigens by the immune system, processing through the cellular antigen processing machinery and presentation by human leukocyte antigen (HLA) class I molecules on the tumor cell surface are essential prerequisites. These HLA class I molecules consist of a heavy chain and a non-covalently bound light chain (encoded by the *Beta-2-microglobulin* [*B2M*] gene), both of which are essential for functional antigen presentation. The likelihood of HLA binding for a defined peptide depends on the HLA genotype, as every individual harbors six alleles (*HLA-A*, *HLA-B*, *HLA-C*, two alleles each) that encode for HLA class I heavy chains[9].

The specific mutational steps required for malignant transformation during the evolution of MSI tumors are thus also responsible for their pronounced immunogenicity. MSI tumors are commonly associated with dense lymphocyte infiltration and pronounced local responses of the adaptive immune system at the tumor site[10–14]. Immune recognition of MSI tumor cells is not only responsible for a comparatively favorable clinical course but also reflected by the high sensitivity of advanced-stage MSI cancers toward immune checkpoint blockade (ICB)[15,16]. However, some patients do not respond to ICB treatment.

We and others previously showed that specific T cell responses against a few MMR deficiency-induced frameshift neoantigens occur prior to and after ICB[15,17,18]. However, the landscape of frameshift peptides and potential epitopes in MSI cancer has not been described systematically. One important reason for this gap of knowledge is the fact that short-read next-generation sequencing (NGS) approaches have a limited sensitivity for the detection of indel mutations at homopolymer sequences such as frameshift peptide-inducing cMS[19–21].

Here we map the frameshift peptide landscape of the two most common MMR-deficient cancer types, CRC and EC, using a tool for the quantification of cMS mutation patterns (ReFrame, REgression-based FRAMEshift quantification algorithm) combined with NetMHCpan 4.0, a state-of-the-art in silico epitope prediction tool[9]. We reveal and functionally validate a set of previously unknown shared frameshift neoantigens in MSI cancers.

Our results provide evidence for continuous immunoediting during MSI tumor evolution and underline the potential of neoantigen-based vaccines against MSI cancers.

## Results

### cMS mutation frequencies in MSI CRC and EC. Short-read NGS approaches are not ideally suited for mutational and

frameshift peptide profiling of MSI cancers[19–21], showing a high variability in mutation frequency regarding the detection of mutations in different cMS candidates like those located in the genes *TGFBR2*, *SLC35F5*, or *TFAM* (Supplementary Table 1). Importantly, cMS repeats of increased length, which are most susceptible to mutations and therefore encompass the most important mutational targets during MMR-deficient tumorigenesis, are missed by NGS technology that is in common use today[3,5,19,22–26].

To fill this gap and precisely quantify cMS mutation patterns and their resulting translational reading frames in MMR-deficient cancers, we developed an algorithm based on fragment length analysis as the current gold standard for the detection of MSI. ReFrame, our REgression-based FRAMEshift quantification algorithm, allows unbiased quantitative detection of indel mutations by solving a linear system of mathematical equations to remove stutter band artifacts, which result from polymerase slippage events during PCR amplification and subsequent nucleotide gains and losses similar to MSI-induced indels (Supplementary Fig. 1).

We used ReFrame in a series of MSI CRCs ($n = 139$; Supplementary Table 2) to screen for mutations in 41 cMS residing in 40 target genes derived from the first comprehensive cMS database (SelTar*base*, Version 201307)[27]. In addition, we investigated mutation profiles in a cohort of MSI ECs ($n = 28$).

In agreement with previous reports[23,27], our results show that the load of indels at cMS in MSI CRC and EC is high and that multiple concomitant indels at several cMS in the same tumor are very common. Although most CRCs and ECs were distinguishable based on the cMS mutation patterns, a large set of cMS mutations were shared by the majority of MSI CRCs and/or MSI ECs (Fig. 1 and Supplementary Fig. 2).

Moreover, we observed a significant variation of the number of mutations per tumor, ranging from 8 to 29 (median: 20) out of 41 analyzed cMS in MSI CRC and from 8 to 25 in MSI EC (median: 18). The observed variation suggests potential differences in the frameshift peptide load of MSI tumors. Potential clinical consequences, e.g., for the sensitivity toward ICB, should be assessed in future clinical studies[28–30].

ReFrame is not only able to quantify mutation frequency but also to distinguish indel mutation types, which is crucial for the prediction of the frame of the resulting frameshift peptides. As the translation of nucleotide into amino acid sequences is based on three base codons, every mutation in a homopolymer region can either result in a simple deletion or insertion of amino acids or in two entirely different frameshift peptide reading frames: deletions of one nucleotide (further referred to as *minus 1* (*m1*)) or insertions of two nucleotides (*plus 2* (*p2*)) will result in a shift to a frame here referred to as minus one (M1), while deletions of two nucleotides (*minus 2* (*m2*)) or insertions of one nucleotide (*plus 1* (*p1*)) will result in a shift to a frame referred to as minus two (M2) (Fig. 2, Supplementary Fig. 3, and Supplementary Table 3). The results demonstrate that *m1* mutations, resulting in M1 reading frames, were the predominant mutation type (77% in MSI CRC, Fig. 2). The M1/M2 distribution varied significantly across distinct cMS, with significantly elevated numbers of M2 mutations in *BANP*, *TAF1B*, and *ELAVL3*, whereas in *ACVR2A*, *HPS1*, *SLC35F5*, and *TCF7L2* there were significantly more mutations leading to an M1 frameshift than expected by chance (Bonferroni corrected binomial test, $p < 0.05$; Supplementary Table 4).

### Landscape of frameshift peptides and predicted major histocompatibility complex (MHC) ligands. Following the detection of shared indel mutations in MSI CRC and EC, we evaluated the

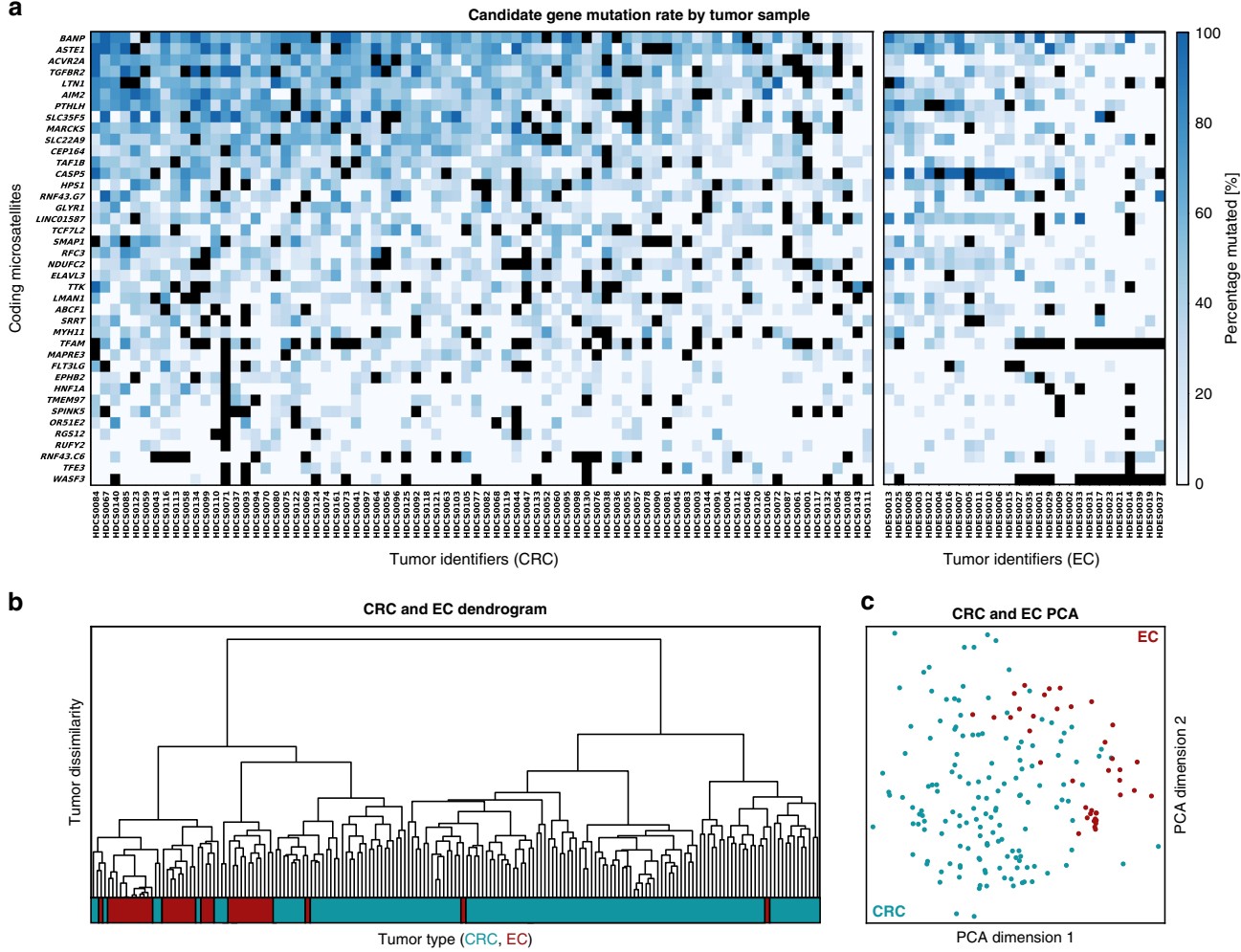

**Fig. 1 Mutation frequencies of cMS in MSI CRC and EC. a** The relative frequency of mutant alleles is shown for 41 cMS (rows) in CRC and EC tumor samples (columns) derived using ReFrame. cMS were sorted top–down according to their mutation frequency indicated by blue boxes of different intensity. Dark blue represents high mutation frequency, whereas pale blue represents low mutation frequency and white absence of mutations. Black boxes indicate missing data points. The respective tumor samples are shown below each column. cMS were analyzed for both CRC and EC and are depicted separately for each tumor type (left panel: CRC, right panel: EC). The complete dataset is provided in Supplementary Fig. 2. Source data for Supplementary Fig. 2 are provided as a Source Data file. **b** Dendrogram of CRC and EC samples with respect to their mutation patterns where the color bar below indicates the tumor type (CRC, blue; EC, red). **c** Principal component analysis of EC (red) and CRC (blue) samples with respect to their mutation patterns.

immunogenic potential of the frameshift peptides and predicted MHC ligands resulting from antigen processing.

We used NetMHCpan 4.0, a state-of-the-art MHC ligand prediction tool based on artificial neural networks, to predict neopeptides that are possibly presented as epitopes by HLA class I antigens encoded by the most important HLA supertypes[9,31,32]. Applying commonly accepted $IC_{50}$ thresholds, we distinguished between three classes of peptides with high ($IC_{50} < 50$ nM), low (50 nM < $IC_{50} < 500$ nM), and very low (500 nM < $IC_{50} < 5000$ nM) predicted HLA-binding affinity[31,33]. As a first step, we analyzed all possible frameshift peptide sequences derived from the M1 and M2 frameshifts of the 41 cMS. We then complemented this set to cover all possible frameshift peptides ($n = 524$) derived from 264 cMS with a length of ≥8 nucleotides published in SelTar*base* (Supplementary Data 1)[27]. Our results indicate multiple frameshift peptides resulting from M1 or M2 frameshift mutations that are potentially recognized by the immune system. We detected a wide range of variability with regard to the number of predicted putative epitopes maximally contained within a defined frameshift peptide. The highest number of predicted putative high-affinity epitopes within a

frameshift peptide was 23 (for the M1 frame of *P4HB*), (low affinity: 92 predicted putative epitopes in M1 *SPINK5*; very low affinity: 375 predicted putative epitopes in M1 *P4HB*). Other cMS mutation-induced frameshift peptides showed a complete lack of the predicted epitopes (Fig. 3 and Supplementary Data 2).

For HLA-A*02:01, the most common HLA allele in the USA European Caucasian population[34], one or more high-affinity peptides were predicted for 19.8% of the frameshift peptides. HLA-A*02:01 epitopes with lower affinity were present in 39.5% (≤500 nM) and 59.7% (≤5000 nM) of candidates (Supplementary Fig. 4 and Supplementary Table 5. Plots depicting predicted HLA-binding peptides for all frameshift peptides and HLA alleles are available in the Source Data folder.).

To make the potential impact of certain cMS candidates more tangible and to identify frameshift peptides with potentially highest relevance for immune recognition, we defined a general epitope likelihood score (GELS; see "Methods" section "Computation of immunological scores"). GELS accounts for MHC ligand prediction and the prevalence of the respective HLA allele in a defined population, as the latter influences the probability of a frameshift peptide to encompass an MHC ligand recognized by

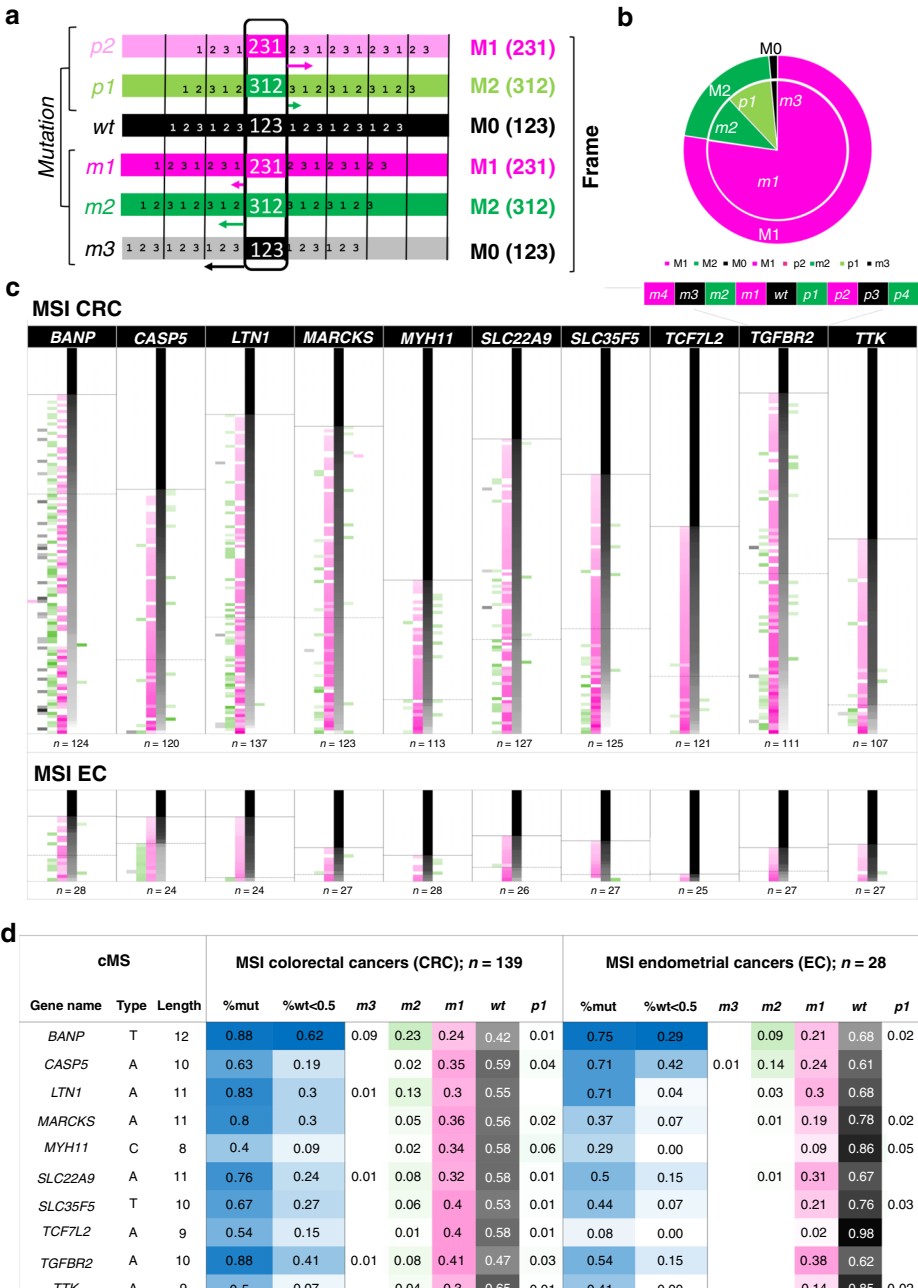

**Fig. 2 Mutational pattern distribution in cMS based on ReFrame analysis. a** Scheme of frameshift mutations (m3, m2, m1, wt, p1, p2) on the left and their corresponding frames (M0, M1, M2) on the right. The numbers 1–3 indicate base triplets in the corresponding frame. Arrows mark the shift between the wt frame and the alternate frame. **b** Distribution of all cMS frameshift mutations (m3–p2) and their corresponding frames (M0, M1, M2) in MSI CRC quantified using ReFrame. Mutations in all MSI CRC samples were classified in corresponding reading frames (M0, black; M1, magenta; M2, green) and their overall allele ratios quantified. **c** The detailed mutational patterns of ten representative cMS are depicted with their respective frequency of mutation for all possible resulting frameshift mutations (m4–p4) in MSI CRC and EC (see Supplementary Fig. 3 for complete dataset). Each row constitutes one analyzed tumor sample with its related allele ratios. For each cMS, tumors were sorted by the proportion of wild-type alleles top to bottom. The number of samples analyzed for a certain candidate is indicated below each candidate's figure. Color indicates the resulting reading frames as in **b**. Intensities represent ReFrame-calculated ratios from white (0%) to full-intensity magenta/green/black (100%) according to the resulting reading frame of the column. The annotated solid lines (first horizontal line top down) show the end of the non-mutated tumor samples (cutoff: 15%), while the dotted lines (second horizontal line top–down) mark the beginning of tumors being >50% mutated, associated with biallelic hits within the respective sample. Source data are available as a Source Data file. **d** Calculated mutation frequencies and mean allele ratios of most common indel mutation types (m3–p1) resulting from ReFrame analysis in ten representative cMS candidates sorted by length. %mut proportion of mutant tumors, %wt<0.5 proportion of tumors with biallelic hits. Columns m3–p1 display average frequencies of the respective indel mutations over all the samples tested.

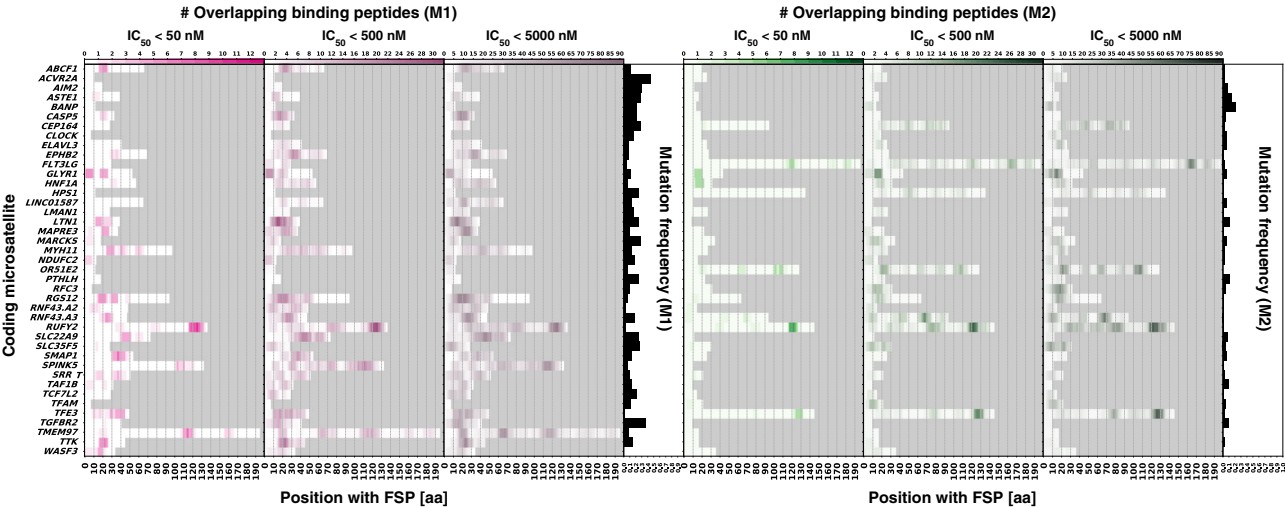

**Fig. 3 MHC ligand predictions for a dataset of 82 analyzed frameshift peptides.** The figures display the predicted epitopes for the M1 and M2 frameshift peptides derived from all 41 cMS candidates over the length of the respective peptide. All epitopes predicted for three binding affinity thresholds (IC$_{50}$ < 50 nM, IC$_{50}$ < 500 nM, and IC$_{50}$ < 5000 nM) are shown for the M1 frameshift peptides (left, magenta) and the M2 frameshift peptides (right, green). All candidates are sorted in alphabetical order with their respective frequency of mutation according to the ReFrame results. The gray field represents background and indicates C-terminal ends of the respective frameshift peptides.

the immune system in a patient of this population[9,34]. We calculated GELS for all frameshift peptides using HLA allele frequencies for USA European Caucasians (calculations for additional ethnic groups are provided in Supplementary Data 3).

Accounting for a potential relation between immunogenicity and mutation frequency, we noticed that the most commonly mutated cMS located in the *ACVR2A* gene showed a very low GELS ($p_{mut}$ = 91%, GELS = 5.1%), whereas very high GELS candidates seemed to be associated with a low mutation frequency (i.e., *TMEM97*, $p_{mut}$ = 27%, GELS = 91.1%; *SPINK5*, $p_{mut}$ = 26%, GELS = 91.1%; *RUFY2*, $p_{mut}$ = 16%, GELS = 90.4%; $p_{binding}$ = 50% in USA European Caucasian population; Supplementary Data 3). Hierarchical clustering of cMS candidates on all tumor samples revealed the existence of three distinct populations of cMS (Fig. 4a), which was retained in *B2M*-wild-type ($n$ = 99) but not in *B2M*-mutant ($n$ = 33) tumors (Fig. 4b).

**Immunoselection during MSI carcinogenesis.** In order to systematically evaluate whether these observations may result from immunoediting, i.e., counterselection of emerging cancer cell clones that harbor highly immunogenic cMS mutations (high GELS frameshift peptides), we analyzed potential differences between the observed and expected distribution of cMS mutations, first in an HLA type-independent approach. Already here we observed a significant inverse correlation between GELS and mutation frequency with Pearson's $r$ = −0.45, $p$ = 0.0078 at $n$ = 41 cMS for endometrial tumors and $r$ = −0.42, $p$ = 0.0149 for colon tumors, with a conservative estimate of predicted HLA-binding probability of $p_{binding}$ = 50%, indicating that a high GELS was related to lower mutation frequency (Fig. 4c). The correlation remained significant even at the lowest epitope fidelity levels of $p_{binding}$ = 10%, with $p$ = 0.0145 for endometrial and $p$ = 0.0031 for colon cancers.

The observation suggests that emerging tumor cell clones with highly immunogenic frameshift peptides are counterselected (Fig. 5), showing that immunoediting may leave its traces in frameshift peptide/cMS mutation patterns in MSI cancers[35–38]. Interestingly, the significant inverse correlation was only detected among *B2M*-wild-type tumors. *B2M*-mutant tumors, in which immune selection on the basis of HLA class I antigen presentation should not apply, only a trend was observed

(Fig. 4d), which possibly reflects effects of immune surveillance prior to *B2M* mutation. To account for confounding factors potentially influencing this observation, we investigated a potential relationship between the cMS length and our observed negative correlation. cMS length is a well-known factor influencing the likelihood of indel mutations on the observed mutation frequency[23,27,39] (Supplementary Fig. 5). However, our analysis demonstrated that GELS was not related to the length of the corresponding microsatellite. Moreover, we replicated the correlation analysis with length-adjusted relative mutation frequencies (relative $p_{mut}$, computed by subtracting the length-specific M1 average mutation frequency from the observed M1 mutation frequency for each microsatellite), the negative correlation between GELS and relative mutation frequency in *B2M*-wild-type tumors was retained (Supplementary Fig. 8a, b).

In order to specifically address HLA-type dependence of immunoediting, HLA-A*02:01 status was determined in MSI CRC and EC[40,41]. HLA-A*02:01 was present in 34 (47.9%) of 71 MSI CRC and 13 (47.1%) of 27 MSI EC samples, in line with proportions reported for German populations (prevalence of HLA-A*02:01: 46.2%, $n$ = 39,689, www.allelefrequencies.net). A significant negative correlation (Pearson's $r$ = −0.15, $p$ < 0.001) was observed between cMS mutation frequency and the likelihood of HLA-A*02:01 ligands resulting from the respective cMS mutation in the HLA-A*02:01-positive but not in the HLA-A*02:01-negative group of MSI CRC (Fig. 5a). Interestingly, this significant HLA-A*02:01-restricted negative correlation was also observed in MSI EC (Fig. 5b), independently supporting the concept of HLA class I-related immunoediting of MSI tumors in the colorectum and endometrium. Again, replication of the analysis using relative mutation frequencies confirmed the negative correlations (Supplementary Fig. 8c).

Despite the statistically significant negative correlation between GELS and mutation frequency, we also observed some outliers (Fig. 4c). We hypothesize that these outliers may reflect distinct effects that potentially influence the probability of a certain cell clone harboring a defined mutation to survive and thrive during tumor evolution. In addition to potential enhancement of immunogenicity, cMS mutations in tumor-suppressor genes are predicted to lead to a growth advantage, at least in cancer or pre-cancer cell clones not directly under

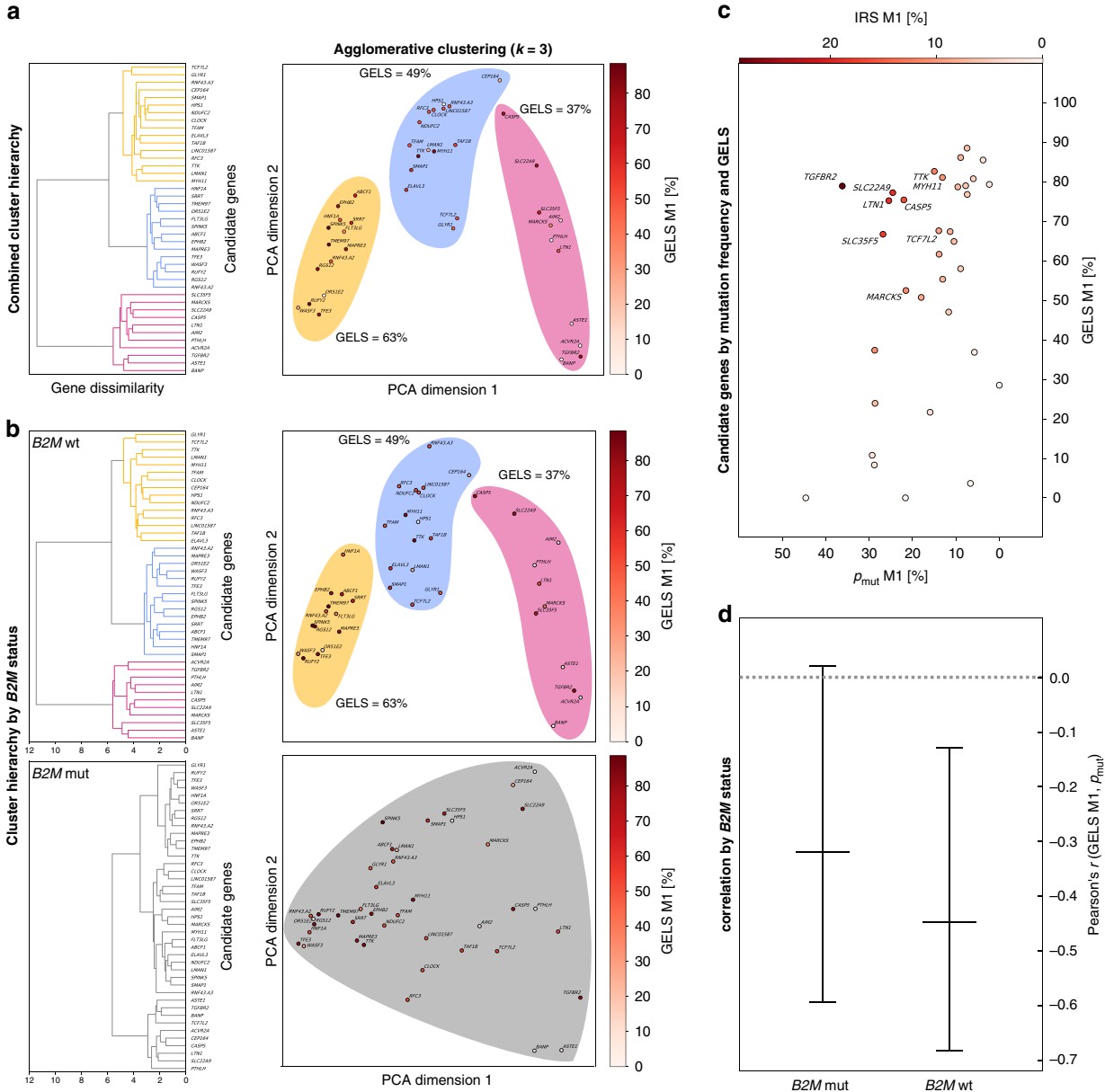

**Fig. 4 Evidence of immune selection from cMS mutation patterns and GELS. a** Hierarchical clustering of cMS candidates on all tumor samples using frameshift abundance features. The full clustering hierarchy is displayed as a dendrogram, showing three clusters (yellow, blue, pink). The same clusters are visualized using RBF kernel PCA with two principal components and colored by their GELS. The three clusters display a trend in their mean GELS, with increasing values from pink to blue to yellow. **b** Hierarchical clustering of cMS on features split by *B2M* status, with wild type at the top and mutated at the bottom. The full hierarchy is displayed as a dendrogram for both feature sets. In contrast to *B2M* wild type, *B2M*-mutated features show no clustering at the computed clustering dissimilarity threshold of 5.7. The same data are again shown using RBF kernel PCA with two principal components. **c** Mutational frequency resulting from one base pair deletions (m1) is shown on the y axis against the GELS of the resulting M1 frameshift peptides (x axis). For GELS, all predicted epitopes ($IC_{50} < 500$ nM) were accounted for, with an assumed probability for a binder to be a true positive of $p_{binding} = 50\%$. Every bubble represents one candidate. The gradient intensity of the bubbles shows IRS, with white color representing a low IRS, while dark red displays a high IRS. All candidates with an IRS of ≥10% are annotated. **d** Correlation between the number of predicted epitopes in cMS mutation-induced frameshift peptides and the frequency of the respective cMS mutations in MSI colorectal cancer separated by *B2M* mutation status. Pearson's r is shown on the y axis, while the different groups of tumors are shown on the x axis. Centers indicate Pearson's r, whiskers indicate 95% confidence intervals. A significant inverse correlation was observed showing $r = -0.42$, $p = 0.0149$ at $n = 41$ candidates for 99 MSI colorectal cancers with wild-type *B2M*, with a conservative estimate of predicted epitope fidelity of $p_{binding} = 50\%$, indicating that high GELS was related to lower mutation frequency (for analysis of relative mutation frequencies, see also Supplementary Fig. 8).

attack of the immune system. Such cMS candidates with high GELS and mutation frequencies should be of great relevance for the interaction between the immune system and MMR-deficient tumor cells. To simultaneously account for mutation frequency and GELS as factors influencing the likelihood of

frameshift peptides being presented to the immune system, we defined an immune relevance score (IRS), which combines GELS with the mutation frequency in tumors computed via ReFrame (see "Methods" section "Computation of immunological scores").

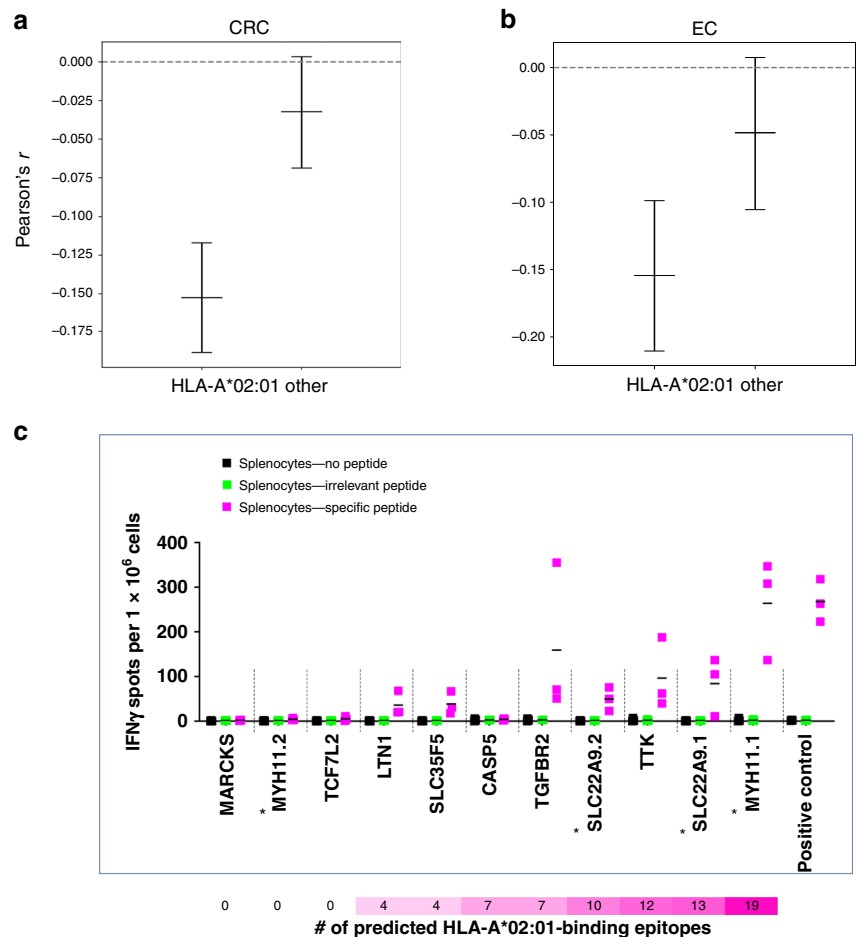

**Fig. 5 HLA-A*02:01-specific analyses.** Correlation between the probability of at least one putative binder truly binding to HLA-A*02:01 within an M1 frameshift peptide and the respective M1 mutation rate in *B2M*-wild-type HLA-A*02:01-positive and HLA-A*02:01-negative CRC (**a**) and EC (**b**). The Pearson's r from the correlation test is shown on the y axis, while the different groups of tumors are shown on the x axis. Centers indicate Pearson's r, whiskers indicate 95% confidence intervals. A highly significant inverse correlation was observed showing $r = -0.15$, $p = 4.4 \times 10^{-16}$ at $n = 34$ HLA-A*02:01-positive CRCs with all parameters kept from Fig. 4d. For $n = 37$ HLA-A*02:01-negative CRCs (other), significance did not persist at $p = 0.29$. Similarly, a significant inverse correlation between HLA-A*02:01-binding scores and mutation frequency was detected for HLA-A*02:01-positive MSI EC ($n = 13$, $r = -0.15$, $p = 0.00002$) but not for HLA-A*02:01-negative MSI EC ($n = 14$) (see also Supplementary Fig. 8c, d). **c** Cellular immune response of M1 frameshift peptides in HLA-A*02:01-expressing A2.DR1 mice. IFNɣ ELISpot was performed with splenocytes from immunized A2.DR1 mice ($n = 3$) re-stimulated with the respective, specific M1 frameshift peptides LTN1, MARCKS, SLC22A9, SLC35F5, MYH11, TTK, TCF7L2, CASP5 (magenta), irrelevant peptide (green), or no peptide (black) as controls. Long frameshift peptides SLC22A9 and MYH were synthesized in two parts (split frameshift peptides are marked by asterisks), with the N-terminal peptide labeled 1 and the C-terminal peptide labeled 2. Each dot represents the number of IFNɣ spots per $1 \times 10^6$ cells for one mouse. Mean values are indicated by horizontal lines. The M1 frameshift peptides are sorted according to the number of predicted HLA-A*02:01 ligands ($IC_{50} < 500$ nm). Source data are available as a Source Data file.

The M1 frameshift neoantigen derived from *TGFBR2*, the first described cMS driver mutation in MSI cancer and also the first ever frameshift neoantigen characterized for its immunological properties in MSI cancer in pioneering studies[18,42,43], displays the highest IRS (28.57%). In addition to this well-characterized frameshift neoantigen, our study uncovered various frameshift peptide candidates with predicted importance for the immune biology of MMR-deficient cancers. The candidates *LTN1*, *SLC22A9*, *SLC35F5*, *CASP5*, *TTK*, *TCF7L2*, *MYH11*, *MARCKS* (all M1), and *BANP* (M2) all displayed an IRS >10% (Fig. 4c and Supplementary Data 3). The spatial distribution of predicted MHC ligands within these high-IRS frameshift peptides is visualized in Supplementary Fig. 6.

To validate the immunogenicity of the predicted HLA-A*02:01-restricted epitopes, HLA-A*02:01-transgenic mice[44] were vaccinated with the respective frameshift peptides. Here interferon gamma (IFNɣ) ELISpot demonstrated reactivity corresponding to HLA-A*02:01 epitope prediction for 8 out of 9 tested candidates, confirming processing and presentation of frameshift peptide-derived HLA-A*02:01 ligands for 6 out of 7 candidates (*LTN1*, *SLC35F5*, *TGFBR2*, *SLC22A9*, *TTK*, *MYH11*; Fig. 5c). For *CASP5*, processing and presentation of HLA-A*02:01-restricted epitopes had been proven previously[45].

Interestingly, candidate genes with a possible tumor-suppressor function were common among the high-IRS genes: *CASP5* (apoptosis induction; IRS: 17.15%), *TTK* (maintenance of chromosomal stability; IRS: 12.38%), *TCF7L2* (beta-catenin signaling; IRS: 11.32%), *MYH11* (cell structure and proliferation; IRS: 11.11%), and *BANP* (migration and invasiveness; IRS: 10.73%) were all previously reported in the literature[46–54]. This observation may suggest that highly immunogenic frameshift peptides are tolerated preferentially if the cells gain a

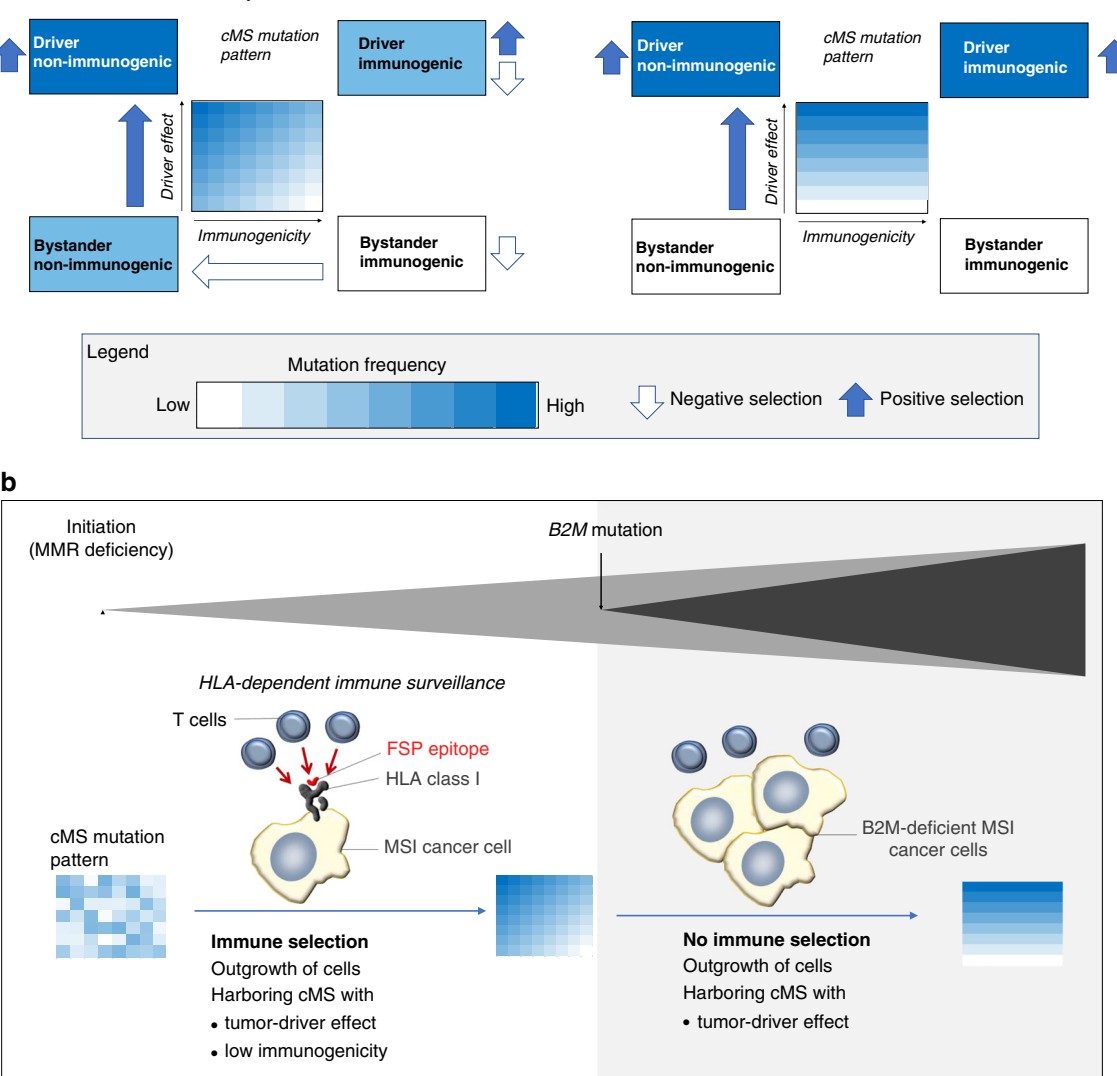

**Fig. 6 Influence of immunoediting on cMS mutation patterns in MSI CRC. a** Coding mononucleotide mutations in MSI cancer can lead to the inactivation of tumor-suppressor genes and therefore have a driver effect on tumor development. Simultaneously, the same mutations can trigger the generation of frameshift peptides with a certain immunogenicity. In the scheme, cMS are located in a two-dimensional lattice depending on the strength of the driver effect resulting from a cMS mutation (y axis) and the immunogenicity of the respective frameshift peptide (x axis). Color illustrates cMS mutation frequency, with dark blue representing high mutation frequency and white absence of mutations (according to heat map in Fig. 1a). The combination of positive selection (blue arrows) and negative selection (white arrows) pressure in *B2M*-wild-type tumors leads to highest frequencies for non-immunogenic driver cMS mutations and lowest frequencies for immunogenic non-driver cMS mutations (left panel). In *B2M*-mutant tumors, negative HLA class I-dependent selection is absent; therefore, expected mutation frequencies are independent from immunogenicity of frameshift peptides. **b** Scheme of MSI CRC evolution. *B2M* mutation can occur as a mechanism of immune evasion during tumor progression. *B2M* mutations lead to a breakdown of HLA class I-mediated antigen processing and therefore interferes with T cell-mediated elimination of MSI cell clones harboring immunogenic frameshift peptides (also see Supplementary Fig. 9).

compensatory survival advantage from the mutation by switching off a tumor-suppressive pathway, supporting their role of propelling MSI tumor evolution (Fig. 6 and Supplementary Fig. 9).

## Discussion

MMR-deficient tumors, due to their well-defined mechanism of genomic instability, represent an ideal tumor type to study the evolution of solid cancer development and the role of the immune system during this process. By analyzing a broad spectrum of cMS-encompassing genes that are susceptible to mutation in

MMR-deficient cells, we were able to identify recurrent mutations and frameshift peptides and to provide first evidence suggesting immunoediting during MSI cancer development.

The results of our study demonstrate that, in contrast to neoantigens in many other cancer types, which typically differ between tumors or even occur as "private" mutational neoantigens, MMR-deficient cancers share a large pool of frameshift peptides: the dominance of alterations of the M1 type resulting from one-base pair deletions (*m1*) emphasizes that MMR-deficient cancers not only share similar sets of genes inactivated by MMR deficiency-induced mutations but also precisely the

same frameshift peptides resulting from these mutations, allowing the definition of a shared frameshift peptide set for MMR-deficient cancers.

Using NetMHCpan 4.0, we predicted a large set of potential MHC ligands in shared frameshift peptides. This number may even increase when using looser prediction thresholds, as recommended in a recent study evaluating the performance of MHC ligand prediction tools[55–57]. Although many frameshift peptides do not encompass such ligands for any of the common HLA types, our calculations demonstrate that the vast majority of MSI cancers are predicted to generate one or more frameshift peptides potentially recognizable by the host's immune system. Validity of positive and negative predictions of HLA-A*02:01 ligands could be proven for eight out of nine shared frameshift peptide candidates. The immunological relevance of mutation-induced frameshift peptides is further supported by the observation of common frameshift peptide-specific T cell responses in patients with MSI cancer and Lynch syndrome mutation carriers[8]. Several of the frameshift peptides derived from common cMS mutations encompass hot spot sequences for which multiple MHC ligands have been predicted (indicated by dark colors in Fig. 3), suggesting that these might be of increased interest for further evaluation[56,57].

By combining quantitative cMS mutation analysis with a specific immune score that accounts for the prevalence of the respective ligand-binding HLA molecules in the population, we identified patterns in the cMS mutation spectrum of MSI cancers suggestive of immune selection: candidates that encompass immunogenic epitopes predicted to bind to common HLA types tend to occur less frequently in manifest MSI cancers. This could be confirmed specifically in HLA-A*02:01-positive patients whose tumors, CRC and EC alike, presented significantly less mutations giving rise to predicted HLA-A*02:01 ligands than expected in a neutral scenario. Our observations support the concept that immune surveillance is a major force shaping the natural course of MMR-deficient cancer development[4,25,26,37,58]. Depletion of expressed neoantigens, similar to what our data suggest, has recently also been reported in lung cancer[59].

Interestingly, mutation patterns suggestive of HLA class I-dependent immunoediting were only observed in B2M-wild-type tumors. This encourages further studies that systematically address the effects of other immune evasion mechanisms such as alterations of HLA class I heavy chains or components of the antigen-processing machinery[37] on antigen landscapes in MSI cancer.

Alternative GELS-independent reasons underlying the observed relations between mutation frequency and predicted immunogenicity do not appear likely. There is no evidence for specific indel mutational signatures[60] favoring the generation of less immunogenic frameshift peptides in MMR-deficient cells, because potential MHC-binding motifs in frameshift peptides are generally located downstream and independent of the actual mutation event. In addition, the median GELS of frameshift peptides resulting from indels affecting A/T repeats (0.617) was virtually identical to the one related to G/C repeats (0.615), further arguing against a signature-related bias. Moreover, an artificial nature of the observed mutations, for example, resulting from formalin fixation, can largely be excluded, because formalin fixation-induced artifacts mostly represent C>T alterations with a limited allele frequency[61], and measures were taken to minimize potential artifacts, including use of buffered formalin, limited fixation times, and preparation of large amounts of template DNA for each tumor.

Another factor potentially influencing the immunological consequences of cMS mutations in MMR-deficient cancers is nonsense-mediated RNA decay (NMD). NMD is known to differentially affect the degradation of target mRNAs in MSI tumors[62–64]. Mutant mRNAs that escape NMD surveillance are expected to account for the high frameshift peptide load in MSI tumor cells. Although NMD may interfere with the translation of mutant mRNAs into frameshift peptides, complete elimination of NMD-sensitive mutant mRNAs by NMD is rare[62–64], and even degradation of NMD-sensitive, mutant mRNAs does not entirely preclude T cell responses specific for the respective neopeptides[65,66]. Systematic studies analyzing the effect of NMD on the antigen spectrum and immune surveillance in MSI tumors are warranted.

Other studies failed to detect evidence for negative selection of immunogenic, neoantigen-inducing mutations in cancer and thereby immunoediting[67,68]. This discrepancy may be related to the fact that the present study addressed frameshift peptides derived from indel mutations, whereas point mutation-induced antigens were examined in the other studies[67,68]. Moreover, ReFrame allows to distinguish between M1 and M2 frames resulting in entirely different frameshift peptides, thus providing the resolution required for distinguishing the entirely different immunological consequences of mutations causing an M1 or M2 frameshift. Our study also accounts for the population frequency of defined HLA types and provides specific information about HLA type of patients studied. In general, the detectability of specific counterselection events is supported by three specific features of MMR deficiency: first, MMR-deficient cancers in contrast to other tumors share precisely the same mutations, because the location of a cMS within a gene determines its susceptibility for indel mutations in MMR-deficient cells; second, MMR-deficient cancers due to the dramatically elevated rate of somatic mutations per cell division are expected to harbor a significantly higher proportion of MMR deficiency-induced mutations compared to age-related mutations that have occurred prior to tumor initiation, thus enhancing the visibility of negative selection events; third, counterselection against frameshift peptides may be particularly pronounced, as MMR deficiency-induced mutations often lead to generation of long frameshift peptides with potentially multiple epitopes, against which no central immune tolerance exists[69].

The observation of HLA-A*02:01-dependent immunoediting during the development of MMR-deficient cancers also implies that a person's HLA genotype may have an influence on the immune environment during MSI tumor evolution. Given the existence of immune-relevant frameshift peptides that may be bound only by a certain type of HLA molecules, it is reasonable to assume that HLA genotype may be a modifier of cancer risk. This may also explain possible variations of Lynch syndrome penetrance or different rates of MMR deficiency previously suspected between distinct populations[70]. Future studies on the natural course of Lynch syndrome should account for this factor.

Our study has the following limitations. The list of frameshift peptides analyzed with ReFrame is not exhaustive. The 40 candidates analyzed in our study are derived from a comprehensive database encompassing 558,000 coding mononucleotide repeats[27], and mutation data provided by SelTarbase.org (Version 201307), which contain information about 4433 distinct mononucleotide repeats from 616 studies, were used for candidate selection. However, additional frameshift mutations resulting from other, predominantly shorter and less frequently mutated cMS can occur in MSI cancers. Because only paraffin-embedded tissue was available from CRC and EC samples, HLA typing could only detect absence or presence of HLA-A*02:01, restricting HLA-related immunoediting analyses to HLA-A*02:01. In addition, we can only propose an atlas of predicted potential neoepitopes in MSI cancers, and a possible influence of NMD could not be systematically addressed. Although previous studies

evaluated a few of the predicted candidates[18,45], supporting the general validity of the in silico predictions, functional validation in our study was only feasible for HLA-A*02:01 ligand predictions due to the availability of a suitable transgenic mouse model[44]. Further studies are encouraged to directly detect frameshift peptide-derived neoepitopes, e.g., by elution from HLA class I complexes on the surface of MSI cancer cells.

The shared frameshift peptide landscape of MSI cancers encourages preventive vaccines, particularly in the setting of Lynch syndrome. If we are able to enhance the abundance of T cells recognizing frameshift neoantigens by a specific vaccine, we may shift the balance toward elimination of emerging cancer cells, thereby reducing the likelihood of escape variants leading to outgrowth of clinically manifest tumors. The safety and immunological efficacy of such a frameshift peptide-based vaccine has already been demonstrated in a first clinical phase I/IIa trial (https://clinicaltrials.gov/show/NCT01461148). If the immune system can be specifically sensitized toward frameshift neoantigens resulting from driver mutations that inactivate tumor-suppressor genes, such as the ones we evaluated in this study, tumor evolution should be influenced in a way that outgrowth of "dangerous" MSI cancer cell clones should become significantly less likely.

In conclusion, frameshift mutation landscapes in MSI cancers suggest negative selection of mutations that give rise to highly immunogenic frameshift neoantigens. This supports the immunoediting concept in non-viral human tumors. Frameshift peptide-based vaccination approaches for the prevention of MMR-deficient cancers should account for the natural immune surveillance during their development and focus on strengthening the host's immune response against neoantigens that are related to essential driver mutation events.

## Methods

**Tumor specimens**. Formalin-fixed, paraffin-embedded (FFPE) archival tissue blocks were collected from 139 MSI CRCs and 28 MSI ECs. Pseudonymized clinical data of each tumor patient is summarized in Supplementary Table 2. Tumors were obtained from the Department of Applied Tumor Biology, University Hospital Heidelberg in frame of the German HNPCC Consortium, the Finnish Lynch syndrome registry, and Leiden University Medical Center. The study was approved by the Institutional Ethics Committee, University Hospital Heidelberg. Informed consent was obtained from all patients.

**Tissue workup and DNA isolation**. FFPE tumor sections (5 μm) were deparaffinized and stained with hematoxylin and eosin according to standard protocols[58]. DNA was isolated from tissue sections after separate microdissection of normal and tumor tissue. Only samples with a tumor cell content of >80% were used for the analysis. Genomic DNA was isolated using the Qiagen DNeasy Tissue Kit (Cat. No. 69506, Qiagen, Hilden, Germany) according to the manufacturer's instructions.

**MSI analysis**. The tumors were characterized for their MSI status using the NCI/ICG-HNPCC five microsatellite marker panel supplemented with additional mononucleotide markers BAT40 and CAT25[71]. Tumors displaying instability in >30% of the analyzed markers were classified as MSI.

**Analysis of frameshift mutations in cMS**. In order to amplify the cMS loci, primers were either obtained from the SelTarbase (Version 201307, www.seltarbase.org)[27] or designed using the primer3 software (Primer3web version 4.0.0, http://primer3.ut.ee/), with one primer of the primer set carrying a 5′ fluorescent (fluorescein isothiocyanate) label. Primers were designed to generate amplicons in range between 100 and 150 nucleotides for robust PCR amplification (Supplementary Table 6). PCR was performed in a total volume of 5 μl containing 0.5 μl 10× reaction buffer (Invitrogen, Karlsruhe, Germany), 1.5 mM MgCl$_2$, 200 mM dNTP mix, 0.3 mM of each primer, 0.1 U Taq DNA polymerase (Invitrogen), and 10 ng of genomic DNA, using the following protocol: initial denaturation at 94 °C for 5 min; 36 cycles of denaturation at 94 °C for 30 s, annealing at 58 °C for 45 s, and primer extension at 72 °C for 1 min; final extension step at 72 °C for 7 min. PCR fragments were separated on an ABI3130xl genetic analyzer (Applied Biosystems, Darmstadt, Germany). Generated raw data were analyzed using the GeneMapper™ Software version 4.0 (ThermoFisher, Waltham, USA). Peak height profiles were extracted and processed using ReFrame based on R version 3.4.3.

**Microsatellite allele distributions analyzed using ReFrame**. In general, PCR amplification of microsatellite loci generates fragments that can vary in length, either due to indel mutations in MMR-deficient cells or due to polymerase slippage during amplification (stutter band artifacts). These two phenomena cause overlays of peak patterns and hamper data interpretation. We developed ReFrame, a REgression-based FRAMEshift quantification algorithm, to allow quantitative analysis of microsatellite mutations by removing stutter band artifacts.

We obtained main peak fractions as a function of microsatellite length $L$, to which a logistic function, in the following referred to as $p(L)$, was then fitted. For each microsatellite in question, an effective length was computed using that fit. We then determined stutter fractions for each gene by calculating the ratios of additional fragments occurring at each microsatellite locus in MMR-proficient control samples ($n = 20$) to establish baseline reference values $p_{ref}$. For each cMS, we computed the expected relative contributions of each indel in the range of $\Delta = -4$ deletion to $\Delta = +4$ insertion to each band in the data as:

$$C^L_{\Delta\Delta'} = \mathbb{I}_{<5}(\Delta' - \Delta) \cdot p_{ref}(\Delta' - \Delta) \cdot \frac{1 - p_{effective}(L, \Delta)}{1 - p_{ref}(0)}, \quad (1)$$

where we defined

$$p_{effective}(L, \Delta) = p(p^{-1}(L) + \Delta), \quad (2)$$

$$\mathbb{I}_{<x}(y) = \begin{cases} 1, & |y| < x \\ 0, & |y| \geq x. \end{cases} \quad (3)$$

We used these relative contributions to set up a linear system for the true peak size without stutter contributions (Supplementary Fig. 1) by requiring

$$C^L \mathbf{p}_{true} = \mathbf{p}_{observed}, \quad (4)$$

$$\min(\mathbf{p}_{true}) \geq 0; \max(\mathbf{p}_{true}) \leq 1, \quad (5)$$

where $\mathbf{p}_{observed}$ and $\mathbf{p}_{true}$ are the observed and true peak sizes, respectively. Resulting allele profiles were imported into a database for further analysis.

Validation of ReFrame was performed in three steps: first, DNA of colonic normal tissue was used to determine baseline deviations of the method in negative controls (Supplementary Fig. 1). In addition, microsatellite-stable cell line DNA (HT29) was used as a control. Finally, three cell line DNAs with differing mutation states (HT29 displaying wild-type peak patterns, LS180 and RKO, displaying mutant peak patterns) were mixed in 10% steps and expected allele distributions were compared to the ReFrame results (Supplementary Figs. 1 and 7).

**Selection of cMS and frameshift peptide sequences**. For MHC ligand prediction, 524 frameshift peptide sequences from 262 mononucleotide changes were retrieved from SelTarbase (Version 201307, www.seltarbase.org)[27]. From an initial series of 558,000 coding mononucleotide repeats, all candidates with a length of at least eight bases were included. In particular cases, other cMS representing putative driver genes, as well as genes which give rise to frameshift peptides with predicted high affinity ligands according to the literature, were also added to the study. In order to also assess potential epitopes located at the junction between N-terminal wild type and C-terminal mutant peptide sequences, the tested peptide sequences all comprised eight wild-type amino acids directly located upstream of the frameshift peptide sequence to encompass possible fusion epitopes. The whole list of used frameshift peptides is given in Supplementary Data 1.

**HLA typing**. Normal tissue DNA isolated from FFPE CRC blocks was genotyped for HLA-A*02:01 status by Sanger sequencing using specific oligonucleotide primers for the amplification of a region spanning the 5′ end of exon 2 of the HLA-A locus with an upstream intronic region. This region encompasses a single-nucleotide variant that allows to distinguish HLA-A*02 from non-HLA-A*02 genes[40] and thus high accuracy classification of samples as HLA-A*02:01 positive or HLA*02:01 negative[72]. For amplification, the following oligonucleotide primers were designed according to a protocol published in ref. [41]: A2-2F: 5′-TCTCAGCCACTCCTCGT C-3′, G-R: 5′-TGTCGAACCGCACGAACTG-3′. After amplification according to ref. [41], Sanger sequencing was performed, and the nucleotide at nucleotide position 78 of the HLA-A gene locus (reference sequence NM_001242758.1) was used for classification, GCTC**Y**CACT with T indicating HLA-A*02:01 and C indicating non-HLA-A*02:01. The method was successfully validated in independent specimens with known HLA type. For B2M mutation analysis, the following oligonucleotide primers were used: exon 1 forward, 5′-GGCATTCCTGAAGCTGACA-3′, exon 1 reverse, 5′-AGAGCGGGAGAGGAAGGAC-3′ (annealing temperature 59 °C); exon 2A forward, 5′-TTTTCCCGATATTCCTCAGGTA-3′, exon 2A reverse, 5′-AAT TCAGTGTAGTACAAGAG-3′ (annealing temperature 57 °C); exon 2B forward, 5′-CATTCAGACTTGTCTTTCAG-3′, exon 2B reverse, 5′-TTTCAGCAGCTTA CAA-3′ (annealing temperature 64 °C)[37].

**MHC ligand predictions**. For MHC ligand prediction, the frameshift peptide sequences derived from each the M1 and M2 mutated alleles were analyzed for the presence of binders using the publicly available prediction tool NetMHCpan 4.0 (www.cbs.dtu.dk/services/NetMHCpan)[9], whose performance has been evaluated to be one of the best of the available tools[55]. As m1- and p2-induced M1 frameshift

peptides (akin to $m2$- and $p1$-induced M2 frameshift peptides) are identical, except for one additional amino acid at the transition between wild type and neo-sequence, we only used M1/$m1$ and M2/$m2$ frameshift peptides for MHC ligand prediction.

Predicted epitopes were subdivided into three classes based on commonly used thresholds. While the first class included epitopes with a predicted affinity of $IC_{50}$ < 50 nM, referred to as high-affinity binders, the second class included all predicted binders <500 nM (low-affinity binders). The last class contained all putative epitopes with <5000 nM affinity (very low-affinity binders). All potential HLA binders with an affinity >5000 nM were discarded. The peptide length of interest was set to 8-mer to 14-mer peptides. A preselection of HLA supertype representatives including HLA-A*01:01, HLA-A*02:01, HLA-A*03:01, HLA-A*24:02, HLA-A*26:01, HLA-B*07:02, HLA-B*08:01, HLA-B*27:05, HLA-B*39:01, HLA-B*40:01, HLA-B*58:01, and HLA-B*15:01 was chosen based on previous recommendations[31,32]. All selected cMS and frameshift peptide sequences were submitted to a Python driver script operating NetMHCpan 4.0[9] to predict putative MHC ligands. The prediction results were processed using a Python script applying the above-mentioned $IC_{50}$ thresholds to all predicted peptides, yielding three datasets of peptides with potential very low, low, and high HLA binding affinity. The resulting datasets were then used to generate figures visualizing the predicted epitopes using matplotlib[73]. To that end, predicted epitopes were counted and mapped for each HLA type, frameshift peptide candidate, and the respective epitope class (high-, low-, or very low-affinity binder). The results of that analysis were used to generate heat maps per candidate and HLA type using another Python script available on github (see below).

**Selection of HLA allele frequency data**. HLA allele frequency datasets were selected from the Allele Frequency Net Database[34] by taking the largest datasets of each ethnicity with at least 10,000 data points and sufficient resolution in HLA alleles. These were further processed together with epitope and mutation data to compute the immunological scores.

**Computation of immunological scores**. For all candidate frameshift peptides, measures of probable immunological relevance were computed based on the above-described predicted $IC_{50}$ values and mutation frequencies. A hierarchy of probabilities for the given candidates to produce immune reactions was computed, including an epitope likelihood score (ELS) per HLA type, a GELS comprising all HLAs under consideration, as well as an IRS. The ELS was defined to describe the probability of a given frameshift neoantigen to be effective across a population, relative to a single HLA:

$$ELS_H(n) = \left(1 - (1 - f_H)^2\right) \cdot \left(1 - \left(1 - p_{\text{binding}}\right)^{|E_H(n)|}\right), \quad (6)$$

where $H \in$ HLA supertypes is a given HLA, $n \in$ cMS is a given frameshift peptide, $f_H$ the allele frequency of a given HLA allele, $p_{\text{binding}}$ the probability, that a given predicted epitope is actually bound, that is the true positive rate of the prediction algorithm, and $E_H(n)$ the set of all epitopes predicted for a given HLA and frameshift peptide. Taken together, $ELS_H$ constitutes the probability of a given candidate $n$ having at least one MHC ligand for an HLA $H$ and a random person from a given population having at least one allele of $H$.

Consequently, the GELS gives the probability of a candidate $n$ having at least one MHC ligand among all HLAs, for which the given HLA is also present in a randomly selected individual:

$$GELS_X(n) = 1 - \prod_{H \in \text{HLA}_X} (1 - ELS_H(n)); X \in \{A, B\}, \quad (7)$$

$$GELS(n) = GELS_A(n) + GELS_B(n) - GELS_A(n) \cdot GELS_B(n), \quad (8)$$

where $\text{HLA}_X$ is the set of HLA types considered for locus $X$.

Finally, the IRS is the joint probability of a given frameshift peptide and its underlying cMS mutation being present in an individual and at least one predicted binder existing for an HLA present in that individual, assuming independence between the presence of HLA alleles and present frameshift peptides:

$$IRS(n) = p_{\text{mut}}(n) \cdot GELS(n). \quad (9)$$

ELS and GELS were computed for all candidate frameshift peptides and HLAs considered using Python on the three output classes of epitope prediction, where binding probabilities $p_{\text{binding}}$ were incremented from 0% to 90% in steps of 10%. HLA allele frequencies were obtained from the Allele Frequency Net Database[34]. IRSs were computed for all candidates with available mutation frequency data.

**Cluster analysis of mutation patterns**. Frameshift mutation abundances ($m4$–$p4$) for each gene and tumor sample were filtered for missing data. For all subsequent clustering experiments, missing values were replaced by the dataset mean. Abundances of frameshift mutations were summarized by their respective reading frame (M2, M1, wt), providing the features used for all subsequent analyses. Resulting features were grouped by tumor sample and candidate cMS, respectively. Hierarchical clustering using Ward's minimum variance linkage[74] was performed for both feature sets grouped by cMS and tumors for all tumor samples considered, as

well as for cMS features considering only *B2M*-wild-type and mutated tumors, respectively. Three clusters of candidate cMS were extracted from hierarchical clustering both for features considering all tumor samples and features considering *B2M*-wild-type tumors only.

**Vaccination of A2.DR1 mice**. A2.DR1 mice, provided by the Institute Pasteur (Paris, France), are transgenic for HLA-A*02:01 and HLA-DRB1*01:01 and lack all murine MHC molecules through knockout of β2m, H-2Db, and the whole MHC class II locus[44]. All animal procedures followed the institutional laboratory animal research guidelines and were approved by the governmental authorities (Regierungspraesidium Karlsruhe, Abteilung 3 - Landwirtschaft, Ländlicher Raum, Veterinär- und Lebensmittelwesen, approval number G302/19).

For the experiments, mice were assigned to the age- and sex-matched groups. The M1 peptides LTN1 KKKMVRLDLLMRYLKAIKRMKNVYLQKERRL, MARCKS SNETPKKKRSAFPSRSLSS, MYH11.1 APGEETRPLSFLLEGLEDVELLKM QMVLRRKKRT, MYH11.2 MVLRRKRTLETQTSMEPRPVNKQLSTVLHHGK, SLC22A9.1 VKGSPSCPLRDLQTLWPILALISMSSIWGT, SLC22A9.2 SMSSIWGT MFSCCRLSLVQSSSWPTVLHLG,

SLC35F5 VAKISFFFALCGFWQICHIKKHFQTHKLL, TCF7L2 CGPCRRKK SAFATYKVKAAASAHPLQMEAY, and p16INK4a LPNAPNSYGRRPIQVMMM GSARVAELL were synthesized by the GMP & T Cell Therapy Unit at German Cancer Research Center Heidelberg and the peptides CASP5 KQLRCWNTWAK MFFMVFLIIWQNTMF and TGFBR2 KCIMKEKKSLVRLSSCVPVALMSAMTTS SSQKNITPAILTCC were synthesized by Genaxxon bioscience GmbH including high-performance liquid chromatography purification and quality control by mass spectrometry. Due to the length of the M1 frameshift peptides, MYH11 and SLC22A9 were synthesized in two parts each (MYH11.1, MYH11.2, SLC22A9.1, and SLC22A9.2). The lyophilized peptides were dissolved in 100% dimethylsulfoxide (SERVA Electrophoresis) at a concentration of 20 mg/ml and stored at −20 °C. Mice were immunized intradermally three times in 7-day intervals. For each immunization, a pool of 3–4 peptides (100 μg each) was injected per group together with 20 μg CpG ODN 1826 (TIB MolBiol) as adjuvants in a phosphate-buffered saline (PBS)-based solution (Thermo Fisher Scientific). Readout experiments were performed 7 days after the last immunization. The p16INK4A peptide was used as positive control for the induction of an HLA-A*02:01-specific, cellular immune response[75].

**IFNγ ELISpot**. The spleen was removed aseptically, pressed through a 40-μm cell strainer (Falcon), treated with Red Blood Cell lysis buffer (Sigma-Aldrich) to lyse erythrocytes, and washed with RPMI Medium 1640 supplemented with 10% fetal bovine serum (FBS), 100 U/ml penicillin, and 0.1 mg/ml streptomycin (all Thermo Fisher Scientific). Ethanol-activated MultiScreenHTS plates (Merck) were coated with purified rat anti-mouse IFNγ antibody (clone R4-6A2, BD Bioscience) in PBS and blocked with RPMI Medium 1640 containing 10% FBS, 100 U/ml penicillin, and 0.1 mg/ml streptomycin. Splenocytes were seeded at $1 \times 10^6$ cells/well and stimulated with 0.1 μg/ml peptide. Concanavalin A (Thermo Fisher Scientific) at the same concentration was used as an assay high control for IFNγ production. After 20 h of incubation, IFNγ-producing cells were detected with a biotinylated rat anti-mouse IFNγ antibody (clone XMG1.2, BD Bioscience), AKP Streptavidin (BD Bioscience), and BCTP/NBT as a substrate (Thermo Fisher Scientific) for color development. The CTL ImmunoSpot Reader (Cellular Technology Ltd) was used for quantification of spots.

**Reporting summary**. Further information on research design is available in the Nature Research Reporting Summary linked to this article.

## Data availability

Source data underlying Figs. 2c and 5c and Supplementary Figs. 2, 3, 5, and 7 are provided as a Source Data file. The Source data folder contains also the plots depicting predicted HLA-binding peptides for all frameshift peptides and HLA alleles (underlying Fig. 3). Mutation data used for candidate selection are available from SelTarbase [www.seltarbase.org]. All remaining relevant data are available in the article, supplementary information, or from the corresponding author upon reasonable request. Source data are provided with this paper.

## Code availability

The source code of all used algorithms can be accessed on GitHub [https://github.com/atb-data/neoantigen-landscape-msi].

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

## Acknowledgements

The present study has been funded in part by grants of the Wilhelm Sander Foundation (Grant number 2016.056.1). The excellent technical assistance of Nina Nelius, Petra Hoefler, and Beate Kuchenbuch is gratefully acknowledged.

## Author contributions

Study conception: A. Ballhausen, M.J.P., M.J., M.K.D., M.K. Molecular analysis: A.B., M.J.P., E.P., F.S., J.W., A.H.S., K.U., M.D., S.K., A.A. Data analysis: A.B., M.J.P., M.J., S.H., E.P., F.S., J.W., A.H.S., K.U., M.D., S.K., A.A., M.S.K., D.H., D.S., M.B., V.H., J.K., A. Benner, A.B.R., M.K. Data interpretation: A. Ballhausen, M.J.P., M.J., S.H., F.S., J.W., A.H.S., K.U., M.S.K., D.H., D.S., J.G., M.B., S.S., H.B., V.H., J.K., A. Benner, A.B.R., M.K.D., M.K. Manuscript writing: A. Ballhausen, M.J.P., M.J., S.H., J.G., M.B., M.v.K.D., M.K. Providing tissue specimens: S.S., H.B., T.S., J.-P.M., S.T.B., M.N., M.v.K.D., M.K. Revision and final approval of the manuscript: all authors.

## Funding

## Competing interests

The authors declare no competing interests.
