## [Peer Review File · Nature Communications]

Reviewers' comments:

Reviewer #2 (Remarks to the Author): Expertise in computational immunogenomics

Predicting MHC binding potential for shared MSI frameshifts across important HLA super types is an important step towards de-personalizing neoantigen directed therapeutics (e.g. enabling neoantigen vaccines and TCR therapies to be mass produced). However, this paper is unfortunately does not do an adequate job justifying several key design choices:

* Why only analyze mononucleotide repeats? One of papers cited simply said "Only mononucleotide repeats were considered since they represent the most simple type of microsatellites". Similarly, the SelTarbase paper said "Mononucleotide repeats (MNRs) seem to represent the most interesting kind of microsatellites" but the rationale wasn't clear to me. By contrast, in "A molecular portrait of microsatellite instability across multiple cancers" they considered a much larger number of longer repeats: "386,396 loci (112,896 mono-, 63,162 di-, 132,117 tri- and 78,221 tetranucleotides)".

* Why only analyze 40 genes? While I'm sure the restriction to coding mononucleotide repeats limits the choice somewhat, 40 is a very small subset of the total number of possible mutated sites that could be considered. For example, "A molecular portrait of microsatellite instability across multiple cancers" (2017) found "16,812 frameshift MSI events across a set of 6,441 coding MS loci spanning 4,898 genes". It would be nice to get a clearer rationale or justification that not much is being missed by ignoring all other genes.

* I don't understand the rationale for predicting MHC binding of mutant peptides against a representative set of HLA types rather than the actual HLA types of the patients in which the mutations were found.

* Why does it matter whether a reading frame is offset by one or two nucleotides? The rationale for analyzing M1 vs. M2 frameshift is not obvious and even less so once the paper starts comparing differences between M1/M2 ratios across genes. What is relevance of these differences?

More broadly, the title and abstract propagate confusion between "predicted MHC ligand" and "neoantigen". The vast majority of predicted MHC ligands will not be found on a cell's surface bound to MHC and even among those bonafide MHC ligands the majority will not be recognized by T cells. A sequence predicted to bind to a Class I MHC molecule has not been established to be an antigen of any kind (neo- or otherwise). I would recommend replacing "FSP neoantigen" with "frameshifted protein sequence" and "epitope" with "predicted MHC ligand" throughout this paper. Alternatively, getting T cells from any of these patients and showing some kind of T cell responses against these sequences would rescue the use of the terms "antigen" and "epitope".

Smaller issues:

* This paper is very heavy on unique jargon and I suggest that the authors should seek to improve readability by shrinking the number of terms they introduce. For example, mx/px are more readable at -x/+x, similarly FSP can simply be written out as frameshift peptide.

* There should be a better explanation for the assumptions underlying M1EXP and M2EXP (the expected number of M1 and M2 frameshifts), since the low p-values primarily reflect a mismatch between that model and reality.

* Change the color scheme of Fig S4 since I first interpreted the gray as the foreground color.

* Inconsistent use of comma vs. period as decimal separator (e.g. comma in Tables S4 and S5)

Reviewer #3 (Remarks to the Author): Expertise in computational immunogenomics

The authors present a survey of the neoantigenic landscape of MSI tumors and the underlying effects of insertion/deletion (indel) mutations at coding microsatellites (cMS) by developing a novel tool for quantification of cMS patterns that may not be readily captured by short read technologies.

In addition to the inhouse software, Reframe, to determine the allelic frequency of these sites, the authors used NetMHCpan 4.0 to determine which of these mutations resulted in a potential neoantigen based on the binding affinity. However, there are some concerns here :

(a) The authors did not use the specific HLA type of each case, and instead used the commonly used HLA A & B allele supertypes. While HLA genotypes may not be clinically available, there are software tools readily available to perform such typing.

(b) In many places in the manuscript , the authors comment about the immunogenicity of the predicted neoantigens and the peptides being potentially recognized by the immune system. However, no functional validation was actually performed to demonstrate the immunogenic potential of the novel peptide sequences or the effect of immunoediting. Infact, there is no data to show what % of these predicted neopeptides result in a "immunogenic" antigen.

(c)Can the authors comment on the effect of mutations in the HLA regions on such neoantigens in the context of high MSI tumors?

(d) Will the sequence data associated with the study be released? if not, I'd like to see a similar analysis on a publically available dataset so the results of the manuscript can be reproduced

Reviewer #4 (Remarks to the Author): Expertise in in MSI cancers

The authors describe a very interesting concept to evaluate neo-epitopes in MSIH cancers

As the authors point out, MSIH cancers are sensitive to immunotherapy, and patients do develop antigen specific t cells (reference 15,17,18).

Here they seek to describe a landscape of coding microsatellites contributing to neo-epitopes in a cohort of 139 colorectal and 14 endometrial cancers.

Their approach is interesting:

MS loci were PCR amplified and sequenced.

They use a new approach, Reframe to delineate coding microsatellite somatic mutations.

They use an existing approach for predicting HLA matched peptide affinities, NetMHCpan 4.0.

Questions:

1) Why not use each patient's HLA type for their mutations?

From the methods, it's not clear why the authors would not use the patient's corresponding HLA type for their given mutations. While these 153 tumors likely have some unique, shared, and common mutations, with variations of immunogenicity for each based on each patient's unique HLA types. The authors appeared to have picked common Supertypes for HLA A, B, C, based on reference 31, 32, which may be for universal epitopes, rather than unique patients that include immuno-editing/selection.

Example, PEPVAC was "optimized for the formulation of multi-epitope vaccines with broad population coverage."

2) ReFRAME.

The authors approach is interesting, the describe in their methods a validation briefly. However, it's still not clear to me how good this is (sensitivity, specificity, what is gold standard, show it). This could be elaborated further because this is the basis for the rest of the paper.

3) How predictive biologically are the "GELS" candidate rankings? Other data sets were peptides have been eluted for example. (Keeping in mind, this is not taking into consideration unique HLA types for each patient).

4) There were groups seen in MSI tumors (Figure 4), but appeared different in B2M mutant tumors. (B2M mutant suggest lack of presentation and negative selection). This is where I believe corresponding HLA types for predicting neo-peptides are critical, especially if one is to look at B2M-mutant as proof of selective pressure.

5) Figures 2,3,4 are very difficulty to follow in terms of take home messages, labeling, and size. These could be re-organized. Figure 1 is clear enough. Figure 5 is a overview and way too complicated.

Point-by-point response to referees' comments

Reviewer #2 (Remarks to the Author): Expertise in computational immunogenomics

Predicting MHC binding potential for shared MSI frameshifts across important HLA super types is an important step towards de-personalizing neoantigen directed therapeutics (e.g. enabling neoantigen vaccines and TCR therapies to be mass produced). However, this paper is unfortunately does not do an adequate job justifying several key design choices:

** Why only analyze mononucleotide repeats? One of papers cited simply said "Only mononucleotide repeats were considered since they represent the most simple type of microsatellites". Similarly, the SelTarbase paper said "Mononucleotide repeats (MNRs) seem to represent the most interesting kind of microsatellites" but the rationale wasn't clear to me. By contrast, in "A molecular portrait of microsatellite instability across multiple cancers" they considered a much larger number of longer repeats: "386,396 loci (112,896 mono-, 63,162 di-, 132,117 tri- and 78,221 tetranucleotides)".*

Authors' response:

We are grateful to the reviewer for this comment and have adapted the manuscript, better explaining the rationale behind focusing on mononucleotide repeats.

Because tetranucleotide, pentanucleotide or more complex repeats are virtually absent in gene-encoding regions (Kloor et al. 2006, PMID 16416315), and because indel mutations affecting trinucleotide repeats do not lead to translational frameshifts, only mononucleotide and dinucleotide repeats theoretically remain a relevant source of frameshift mutations. Dinucleotide repeats in gene-encoding regions of the human genome, however, are rare and much shorter than non-coding dinucleotide repeats, possibly because of their incompatibility with biologically functional amino acid sequence stretches. Therefore, coding dinucleotide repeats remain generally stable and unaffected by indel mutation events even in MMR-deficient cells (Woerner et al. 2001, PMID 11391615). Known coding dinucleotide repeat mutations in MMR-deficient cancers mostly represent private mutations, with the notable exception of the (CT)₄TTCT repeat in exon 1 of the Beta-2-microglobulin gene (Kloor et al. 2007, PMID 17373663), and therefore are numerically irrelevant and not suitable candidates as a source for shared neoantigens in MSI cancer.

The mentioned study by Cortes-Ciriano (2017, PMID 28585546) further confirms that indel mutations at coding microsatellites are restricted to mononucleotide repeats (Supplementary Figure 3).

* Why only analyze 40 genes? While I'm sure the restriction to coding mononucleotide repeats limits the choice somewhat, 40 is a very small subset of the total number of possible mutated sites that could be considered. For example, "A molecular portrait of microsatellite instability across multiple cancers" (2017) found "16,812 frameshift MSI events across a set of 6,441 coding MS loci spanning 4,898 genes". It would be nice to get a clearer rationale or justification that not much is being missed by ignoring all other genes.

Authors' response:

We fully agree that the restriction to 40 genes is a limitation of our study. However, the selection of the 40 candidates was not random, but the result of *in silico* studies based on a comprehensive and exhaustive genome-wide dataset of 26.8 Mio transcribed mononucleotide repeats of a length of more than 4 nucleotides in the human genome, encompassing 558,000 coding mononucleotide repeats, 874,000 in untranslated 5' or 3' regions, 3,700 in non-coding RNA, and 25,400,000 in intronic pre-mRNA regions (Woerner et al. Nucl Acids Res 2010, PMID 19820113).

To identify coding microsatellites with a high likelihood of recurrent mutation events, mutation data provided by seltarbase.org (release 201307), which contain information about 4433 distinct mononucleotide repeats from 616 studies, were used to identify candidates for ReFrame analysis. We are grateful to the reviewer and have added a statement to the respective section of the Materials and Methods part (page 14).

Though it is very likely that certain neoantigens are missed by focusing on the candidate list of 40 genes (as stated in the discussion, page 11, paragraph 4), there is good evidence that our *in silico* preselection covers the most common immunologically relevant frameshift peptides in MMR-deficient cancer. In fact, the mentioned study by Cortes-Ciriano (2017, reference 22) reports evidence for significant enrichment of mutations only for a small proportion of microsatellites, with a large overlap of microsatellites contained in both Cortes-Ciriano (Supplementary Figure 3) and our study.

ReFrame, a fragment length analysis-based MSI quantification approach, is clearly not a high throughput method and therefore restricts the number of possible candidates tested. However, ReFrame provides two significant advantages compared to previous next generation sequencing studies:

First, ReFrame offers a very high read depth per analyzed marker (theoretically up to 2.7×10^{11} in 38 PCR cycles). Therefore, it is significantly more sensitive and specific for detecting indel events at long mononucleotide repeats than short-read next gen sequencing approaches. This is highlighted by the under-estimation of mutation frequencies in common next gen data-based repositories and publications (see revised Supplementary Table 1).

Second, ReFrame has for the first time provided a frame-specific resolution of indel mutations, allowing the differentiation between M1 and M2 mutations (see response below), sufficiently high for the detection of specific negative selection events during MSI tumor evolution. By using this approach, we were able to reveal significant effects of immunoediting, which had previously gone unrecognized in studies with a larger scale design, but lower target-specific resolution.

** I don't understand the rationale for predicting MHC binding of mutant peptides against a representative set of HLA types rather than the actual HLA types of the patients in which the mutations were found.*

Authors' response:

One of the major aims of the study was the identification of shared neoantigens with the potential of being generalized as immunologically relevant candidates and possible targets for vaccination. Therefore, we predicted MHC binding against a representative set of HLA types in different populations. In order to predict the potential coverage of frameshift peptide vaccines, we provide predictions for common HLA supertypes in distinct populations and provide likelihood scores for frameshift peptides being relevant antigens for all those scenarios (Data S3).

We fully agree with the reviewer that using actual HLA types is better suited for detecting immunoediting and negative selection of immunogenic candidates. Following this suggestion and the suggestions by the other reviewers, we have performed HLA type analysis in the tumor collection of our study.

As full high-resolution HLA genotyping is technically difficult in formalin-fixed, paraffin-embedded tissue specimens and failed in 4 out of 6 specimens tested, we focused on the presence or absence of the HLA-A*02:01 gene as the most common HLA gene in the Caucasian population using published methods (Song et al. 2013, PMID 23954948; Villabona et al. 2014, PMID 24504073). By using this approach, we were able to demonstrate a significant negative correlation between mutation frequency and the likelihood of HLA-A*02:01-specific epitopes resulting from the respective mutation in the HLA-A*02:01-positive, but not in the HLA-A*02:01-negative group. Respective sections have been added to the manuscript, and the correlation analysis was incorporated in Figure 5A. Independent from the observation in MSI CRC, the expanded mutation data set on MSI EC also allowed studying potential immunoediting in endometrial tumors, revealing a statistically significant negative correlation similar to the observation in MSI CRC.

** Why does it matter whether a reading frame is offset by one or two nucleotides? The rationale for analyzing M1 vs. M2 frameshift is not obvious and even less so once the paper starts comparing differences between M1/M2 ratios across genes. What is relevance of these differences?*

Authors' response:

As shifts of the reading frame by one or two nucleotides result in entirely different neopeptide sequences and possible neoantigens, it was essential to evaluate M1 and M2 mutations separately. For example, TGFBR2 M1 mutations result in a neopeptide sequence reading

“SLVRLSSCVPVVALMSAMTTSSSQKNITPAILTCC”, whereas TGFBR2 M2 mutations result in a much shorter neopeptide sequence consisting of the two amino acids “AW”.

For both, analyzing immunoediting and developing preventive vaccines, the differentiation of M1 and M2 frameshift peptides is required. Only by separating M1 from M2 mutations using ReFrame, we were able to detect significant traces of immunoediting in mutation patterns of manifest tumors (see also Main text, page 7, last paragraph, and Figure 2A).

More broadly, the title and abstract propagate confusion between “predicted MHC ligand” and “neoantigen”. The vast majority of predicted MHC ligands will not be found on a cell’s surface bound to MHC and even among those bonafide MHC ligands the majority will not be recognized by T cells. A sequence predicted to bind to a Class I MHC molecule has not been established to be an antigen of any kind (neo- or otherwise). I would recommend replacing “FSP neoantigen” with “frameshifted protein sequence” and “epitope” with “predicted MHC ligand” throughout this paper. Alternatively, getting T cells from any of these patients and showing some kind of T cell responses against these sequences would rescue the use of the terms “antigen” and “epitope”.

Authors’ response:

We are very grateful for this suggestion. We have revised the nomenclature throughout the manuscript according to the reviewer’s suggestion, referring to neoantigens and epitopes only for candidates with proven capability of triggering T cell responses.

Similarly, we have adapted the title, now reading “The shared frameshift mutation landscape of MSI cancers reflects immunoediting during tumor evolution”.

Smaller issues:

** This paper is very heavy on unique jargon and I suggest that the authors should seek to improve readability by shrinking the number of terms they introduce. For example, mx/px are more readable at -x/+x, similarly FSP can simply be written out as frameshift peptide.*

We have replaced unnecessary abbreviations according to the reviewer’s suggestion, now spelling out “frameshift peptide” and other avoidable abbreviations.

We would prefer, however, to keep the nomenclature M1/M2 and m2/m1/wt/p1/p2, as it allows specifically differentiating between indel mutation types and the resulting translational reading frames.

** There should be a better explanation for the assumptions underlying M1EXP and M2EXP (the expected number of M1 and M2 frameshifts), since the low p-values primarily reflect a mismatch between that model and reality.*

Authors’ response:

Thank you for this comment. Significant p values indicate that the observed distribution of M1 and M2 mutations is differing from the assumption of equal probabilities for M1 and M2 mutations. A respective statement has been added to the Table legend, reading “Significant

differences indicate a deviation from a distribution that assumes identical probabilities for M1 and M2 mutations.”

From the perspective of developing a preventive vaccine, this is an important observation, as it points towards a predominance of m1 mutations giving rise to M1 frameshift peptides, which due to their predominance should represent most promising vaccine targets.

** Change the color scheme of Fig S4 since I first interpreted the gray as the foreground color.*

Authors' response:

We would prefer keeping the gradient from white to magenta/green for the predicted MHC ligands as the visible range is larger than in an inverted scheme. To clarify the color scheme, we have added a statement to the figure legend (Supplementary Materials, page 6).

** Inconsistent use of comma vs. period as decimal separator (e.g. comma in Tables S4 and S5)*

Authors' response:

We highly appreciate the thorough reading and have replaced Tables S4, S5 and Supplementary Figure S1 by corrected versions.

--

Reviewer #3 (Remarks to the Author): Expertise in computational immunogenomics

The authors present a survey of the neoantigenic landscape of MSI tumors and the underlying effects of insertion/deletion (indel) mutations at coding microsatellites (cMS) by developing a novel tool for quantification of cMS patterns that may not be readily captured by short read technologies.

*In addition to the inhouse software, Reframe, to determine the allelic frequency of these sites, the authors used NetMHCpan 4.0 to determine which of these mutations resulted in a potential neoantigen based on the binding affinity. However, there are some concerns here :
(a) The authors did not use the specific HLA type of each case, and instead used the commonly used HLA A & B allele supertypes. While HLA genotypes may not be clinically available, there are software tools readily available to perform such typing.*

Authors' response:

We fully agree with this important suggestion. HLA type information has not been clinically available, therefore, we have performed HLA typing for all samples, from which sufficient amounts of material could be obtained. Due to methodological restrictions related to FFPE tissue specimens, we had to restrict the analysis to the HLA-A locus, using previously published methods suitable for fragmented DNA.

By obtaining information about the HLA-A*02:01 status of the respective patients, we were able to detect a significant negative relation between cMS mutation frequency and the probability of HLA-A*02:01-binding peptides in the resulting frameshift sequence ($p < 0.02$, Figure 5A). This

negative correlation was restricted to tumors from HLA-A*02:01-positive patients and interestingly observed independently for MSI colorectal and endometrial tumors. We are very grateful for suggesting HLA typing, which helped to independently validate the hypothesis of HLA-dependent immunoediting and negative selection of immunogenic mutations in our dataset.

(b) In many places in the manuscript, the authors comment about the immunogenicity of the predicted neoantigens and the peptides being potentially recognized by the immune system. However, no functional validation was actually performed to demonstrate the immunogenic potential of the novel peptide sequences or the effect of immunoediting. Infact, there is no data to show what % of these predicted neopeptides result in a "immunogenic" antigen.

We highly appreciate this suggestion. In fact, given the uncertainty associated with neural network-based predictions of MHC binding, functional validation is very important. In order to address this point, we have vaccinated HLA-A*0201/DRB1*0101-transgenic mice with 9 peptides identified as most promising candidates by ReFrame mutation analysis. From these 9 peptides, 7 encompassed predicted potential HLA-A*02:01-restricted epitopes, whereas no such epitopes were predicted for the remaining two. ELISpot analysis from vaccinated transgenic mice revealed T cell immune responses significantly above background for 6 of the 7 candidates with predicted HLA-A*02:01 epitopes (LTN1, SLC35F5, TGFBR2, SLC22A9, TTK, MYH11), whereas none of the other two candidates triggered T cell immune responses. Hence, functional validation confirmed MHC binding predictions for HLA-A*02:01 in 8 out of 9 cases, supporting previous publications that reported immunogenicity for some of the candidate frameshift peptides. The only discrepancy was observed for CASP5, which had previously been shown to contain an immunogenic epitope that can be processed and presented (Schwitalle et al. 2004, PMID 15563124). We have added new sections to the manuscript and a new Figure displaying the results of the ELISpot evaluation of vaccinated mice (revised Figure 5C).

(c) Can the authors comment on the effect of mutations in the HLA regions on such neoantigens in the context of high MSI tumors?

Mutations affecting functional components of the HLA class I-mediated antigen processing and presentation machinery can interfere with the presentation of potential neoantigens to the immune system. In fact, the significant negative correlation between immunogenicity scores and mutation frequency was absent in the group of B2M-mutant MSI CRC (Fig. 4). This point was missing from the discussion section. We have added a respective statement on page 12, reading "HLA class I-dependent immunoediting required the presence of wild type B2M... ."

(d) Will the sequence data associated with the study be released? if not, I'd like to see a similar analysis on a publically available dataset so the results of the manuscript can be reproduced

The ReFrame analysis is based on fragment length analysis data. Results of all fragment length analysis data, on which the study is based, have been released and uploaded to github under the following link:

<https://github.com/atb-data/neoantigen-landscape-msi/tree/master/testfiles>

As described above, validation on publicly available data is not possible, because of (A) the limited sensitivity of short read next gen sequencing methods and (2) the lack of M1/M2 frame-specific mutation data. However, we fully agree that prospective validation in independent cohorts will be very interesting.

Reviewer #4 (Remarks to the Author): Expertise in in MSI cancers

The authors describe a very interesting concept to evaluate neo-epitopes in MSIH cancers

As the authors point out, MSIH cancers are sensitive to immunotherapy, and patients do develop antigen specific t cells (reference 15,17,18).

Here they seek to describe a landscape of coding microsatellites contributing to neo-epitopes in a cohort of 139 colorectal and 14 endometrial cancers.

Their approach is interesting:

MS loci were PCR amplified and sequenced.

They use a new approach, Reframe to delineate coding microsatellite somatic mutations.

They use an existing approach for predicting HLA matched peptide affinities, NetMHCpan 4.0.

Questions:

1) Why not use each patient's HLA type for their mutations?

From the methods, it's not clear why the authors would not use the patient's corresponding HLA type for their given mutations. While these 153 tumors likely have some unique, shared, and common mutations, with variations of immunogenicity for each based on each patient's unique HLA types. The authors appeared to have picked common Supertypes for HLA A, B, C, based on reference 31, 32, which may be for universal epitopes, rather than unique patients that include immuno-editing/selection.

Example, PEPVAC was "optimized for the formulation of multi-epitope vaccines with broad population coverage."

Authors' response:

Using HLA supertypes is essential for identifying vaccine candidates with general applicability in a given population. However, we fully agree that this approach is not ideal for assessing immunoediting. We are very grateful to the reviewers for clearly pointing out the need for

obtaining HLA type information for the patients whose tumors were analyzed in the present study.

As described above, we have now performed HLA typing instead of using general HLA type information for the immunoediting part. Adding HLA type information significantly enhanced the clarity of the obtained results. Incorporating HLA type information provided independent evidence for immunoediting during MSI tumor evolution, for the first time also demonstrating antigen-related immunoediting in MSI endometrial tumors.

2) ReFRAME.

The authors approach is interesting, they describe in their methods a validation briefly. However, it's still not clear to me how good this is (sensitivity, specificity, what is gold standard, show it). This could be elaborated further because this is the basis for the rest of the paper.

Authors' response:

More validation data have been added to the supplementary material as Supplementary Figure 7. These contain experiments, in which DNA of the MSI cancer cell line RKO has been mixed with DNA of the microsatellite-stable cancer cell line HT29. For readout, three markers are displayed representatively to illustrate different scenarios, for which ReFrame was designed: Quantitative differentiation of (1) m1 from wild type alleles, (2) m2 and wild type alleles, and (3) simultaneously m2, m1, and wild type alleles. In spite of using very low amounts of template DNA to simulate difficult amplification conditions, reproducible results with deviations lower than 10% regarding the detected allele frequencies were obtained.

Fragment length analysis still represents a commonly accepted gold standard for MSI analysis, however it is not quantitative. As ReFrame represents the first quantitative approach to detect frame-specific microsatellite mutations, direct comparison to an existing gold standard is not feasible.

3) How predictive biologically are the "GELS" candidate rankings? Other data sets were peptides have been eluted for example. (Keeping in mind, this is not taking into consideration unique HLA types for each patient).

Authors' response:

In order to substantiate the prediction-based rankings, we have performed in vivo validation of HLA-A2 binding predictions for 9 M1 frameshift peptide candidates, confirming the predictions for 8 out of 9.

Further general evidence for the immunogenicity of frameshift peptides in MMR-deficient cancer is provided by previous studies demonstrating FSP neoantigen-specific immune responses in patients with MSI cancer and Lynch syndrome mutation carriers.

However, direct evidence for processing and presentation of MMR deficiency-induced frameshift peptides is lacking so far. We fully agree that such information would be of high

significance and add to our understanding about the frameshift peptide spectrum generated in MMR-deficient cancer cells. We have initiated a collaborative project aiming at the isolation of MHC binding peptides from MSI cancer cells. A statement declaring the importance of such future studies has been added to the discussion: “Further studies are encouraged to directly detect frameshift peptide-derived neoepitopes, e.g. by elution from HLA class I complexes on the surface of MSI cancer cells.”

4) There were groups seen in MSI tumors (Figure 4), but appeared different in B2M mutant tumors. (B2M mutant suggest lack of presentation and negative selection). This is where I believe corresponding HLA types for predicting neo-peptides are critical, especially if one is to look at B2M-mutant as proof of selective pressure.

Authors' response:

We completely agree and appreciate the suggestion. In fact, HLA typing enabled assessing tumors from HLA-A*02:01-positive and HLA-A*02:01-negative patients separately, thus revealing significant negative correlations further supporting the hypothesis that immunoediting in MSI CRC evolution requires the presence of functional HLA class I complexes of the right HLA gene.

The new data have been added as Figure 5A and 5B, indicating significant correlations for the HLA-A*02:01-positive MSI CRC and MSI EC patient groups.

5) Figures 2,3,4 are very difficulty to follow in terms of take home messages, labeling, and size. These could be re-organized. Figure 1 is clear enough. Figure 5 is a overview and way too complicated.

Authors' response:

We are grateful for the comment and tried to enhance the clarity of the presentation. Generally, we have revised the figure legends to make them more concise and easier to follow. Specifically, we have split Figure 2 and present the data previously shown as Fig. 2D now as Table 1. Font sizes and labeling was adapted in Fig. 3, incorporating the nomenclature suggestions mentioned above.

We have replaced figure 5 by a simplified version. The original version of Figure 5, which represented a general overview figure about the study design and results, has been moved to the Supplementary Material (Fig. S8).

Additional comments:

HLA typing has been performed by Johannes Witt. Functional validation in mice has been performed by Alejandro Hernandez Sanchez and Katharina Urban. These authors have been added to the author list. Florian Seidler and Pauline Pfuderer performed additional ReFrame

cMS typing and validation experiments. The authors list has been updated accordingly.

The title has been changed to “The shared frameshift mutation landscape of MSI cancers reflects immunoediting during tumor evolution” following the suggestions made by Reviewer #2.

REVIEWER COMMENTS

Reviewer #3 (Remarks to the Author):

The authors have adequately addressed my comments. The inclusion of HLA typing (though limited to the A allele), as well as functional validation by the authors have significantly improved the quality of the manuscript. I expect the revised manuscript will be a valuable resource for the immunogenomics community. Thank you!

Reviewer #5 (Remarks to the Author):

The study of Ballhausen et al. describes the frameshift mutation landscape of MSI colorectal and uterine cancer, based on a newly developed tool, called ReFrame. The authors find a large set of these mutations to be shared by the majority of investigated MSI cancers. They then predict neoantigens based on the peptides HLA affinities and find a negative correlation between the mutation frequency and the neoantigen likelihood, which is interpreted as immunoediting.

As the authors mention in the discussion section, there has been some recent debate about the evidence of immunoediting and more specifically neoantigen depletion in primary cancer genomics data. Initial studies found strong evidence for this specific form of negative selection (e.g. Rooney et al 2015, Zapata et al 2018), while others didn't and demonstrated important influences of prior mutation probabilities derived from mutational signatures (Martincorena et al 2017, Van den Eynden et al 2019). In this regard, there is also raising doubt on the suitability of HLA affinities to predict neoantigens, especially when studying immunoediting. These studies mainly focused on point mutations, rather than indels. It is currently an open question whether neoantigens generated by these indels are indeed subject to negative immune selection or not.

The manuscript of Ballhausen tackles this problem. It is clearly written, easy to follow, to the point and sustained with informative figures. I was also pleased to see code and data being made available on github. I have some concerns though, that I feel should be addressed before the manuscript can be considered for publication.

1) Most importantly, I'm not entirely convinced about the evidence that the author's findings are indeed explained by immunoediting. While I agree that the negative correlation between mutation frequency and GELS shown in fig. 4C is suggestive, in my opinion this is also the only real evidence in the manuscript. Couldn't there be alternative explanation for this correlation?

- The association with other immune evasion mechanisms might provide some additional evidence, but the authors only looked at B2M mutations, and even there, a trend can still be observed. B2M mutations are also known to be quite rare so the question is whether this association can be made for this small subgroup anyway (I couldn't find the B2M mutation frequency in the manuscript BTW, can the authors provide this?). In this regard, I feel the authors should consider looking at associations with other, more common, immune evasion mechanisms such as HLA LOH and PD1/PDL1 amplifications. This would strengthen their claim about immunoediting.
- Detecting negative selection boils down to detecting fewer mutations than expected, so an accurate expectation model is always crucial (e.g. metrics like dN/dS all have an expected number in the denominator). As far as I can judge, this is absent in the manuscript. In my understanding, the only underlying expectation is that there should not be a correlation as described in fig. 4C? In their response to reviewer 2 the authors indicate that the reason for not using trinucleotide repeats (or in frame indels) is that they do not lead to frameshifts. Would this be a perfect negative control for the immunoediting claim?
- As indicated higher it has been shown that HLA affinities are not independent from mutational signatures (i.e. certain mutations influence specific codons, associated to higher/lower HLA affinities). Wouldn't it be possible that specific indel (ID) signatures result in higher mutation loads in genomic regions encoding non-binding peptides? Showing that this is not the case would again

sustain the authors' immunoediting interpretation.

I understand that new approaches might be hard to implement at this stage of the manuscript, but I feel these concerns should minimally be addressed in the discussion section. I would also be more prudent with the claim that the findings are related to immunoediting (especially in the title) without further evidence.

2) While some validation was added on cell lines, I still miss some benchmarking for the ReFrame tool. How does the tool perform compared to NGS with state-of-the-art indel callings? Is the high indel frequency realistic? Is there a possibility that FFPE somehow influenced this frequency? (As the authors indicate in their responses to all reviewers, HLA typing was not possible on FFPE tissues, so I assume there were DNA quality problems?)

3) The authors claim that their findings are HLA type-dependent, so I expected an HLA genotyping for all patients. I understand from the responses to all 3 other reviewers that this was not technically feasible. I think this should also be added to study limitations in the discussion section and more importantly, any claims that the findings are HLA type-specific should be avoided as they are misleading. Showing a correlation for the most common HLA0201 cannot be generalized in this way. In this regard, I wonder whether the fact that HLA0201 is one of the strongest peptide binding alleles might have had an influence on the results?

Some other, minor comments:

- NMD (nonsense-mediated mRNA decay) is expected to act on indels and I was surprised that this was not addressed in the manuscript. It might actually offer an explanation on why some TSGs behave as outliers (e.g. no expression implies no immune selection).
- Immunoediting or immuno-editing? The authors use both ways of writing.
- Why is the neo in neoantigens italicized in the manuscript?
- Gene names should be italicised.
- Beginning of last paragraph p. 7. "... to distinguish mutation types ...". I was initially confused, thinking about mutation variants (e.g. missense mutations). It gets clearer later but talking about indel mutation types might avoid some confusion for the reader.
- Figure 3. It was not immediately clear to me what the difference between the left/middle/right figures in each panel was. I would add some label to clarify, or even showing only Kd 50nM given the redundancy between the figures.
- An HLA affinity Kd cut-off of 50 nM or 500nM is commonly used. I haven't seen 5000nM before and don't think this is too useful.
- When mentioning in the discussion that other studies (refs 60,61) failed to detect negative selection and immunoediting, the authors' interpretation is superior resolution of ReFrame. I would like to note that most previous studies only focused on single nucleotide substitutions, so I don't agree completely on the interpretation.
- Several choices became clear to me after reading the rebuttal but not the manuscript. I think the authors should try to clarify these in the manuscript as well (e.g. they respond to reviewer 2 that 40 genes is a limitation of the study, but this is not mentioned when discussing study limitations in discussion; same for technical difficulties with HLA typing as I mentioned higher)

Point-by-point response to reviewers' comments

Reviewer #3 (Remarks to the Author):

The authors have adequately addressed my comments. The inclusion of HLA typing (though limited to the A allele), as well as functional validation by the authors have significantly improved the quality of the manuscript. I expect the revised manuscript will be a valuable resource for the immunogenomics community. Thank you!

Authors' reply: We are grateful for the reviewer's comment.

Reviewer #5 (Remarks to the Author):

The study of Ballhausen et al. describes the frameshift mutation landscape of MSI colorectal and uterine cancer, based on a newly developed tool, called ReFrame. The authors find a large set of these mutations to be shared by the majority of investigated MSI cancers. They then predict neoantigens based on the peptides HLA affinities and find a negative correlation between the mutation frequency and the neoantigen likelihood, which is interpreted as immunoediting.

As the authors mention in the discussion section, there has been some recent debate about the evidence of immunoediting and more specifically neoantigen depletion in primary cancer genomics data. Initial studies found strong evidence for this specific form of negative selection (e.g. Rooney et al 2015, Zapata et al 2018), while others didn't and demonstrated important influences of prior mutation probabilities derived from mutational signatures (Martincorena et al 2017, Van den Eynden et al 2019). In this regard, there is also raising doubt on the suitability of HLA affinities to predict neoantigens, especially when studying immunoediting. These studies mainly focused on point mutations, rather than indels. It is currently an open question whether neoantigens generated by these indels are indeed subject to negative immune selection or not.

The manuscript of Ballhausen tackles this problem. It is clearly written, easy to follow, to the point and sustained with informative figures. I was also pleased to see code and data being made available on github. I have some concerns though, that I feel should be addressed before the manuscript can be considered for publication.

1) Most importantly, I'm not entirely convinced about the evidence that the author's findings are indeed explained by immunoediting. While I agree that the negative correlation between mutation frequency and GELS shown in fig. 4C is suggestive, in my opinion this is also the only real evidence in the manuscript. Couldn't there be alternative explanation for this correlation?

The reviewer points out a critical question. We are grateful for the opportunity to discuss potential evidence for immunoediting in more detail.

Our conclusion is based on four major observations:

First, as mentioned by the reviewer, a negative correlation was observed between the epitope likelihood score over all HLA supertypes and the mutation frequency.

Second, a significant negative correlation was observed for predicted HLA-A*02:01 epitopes only in tumors from HLA-A*02:01-positive patients. Third, this HLA-A*02:01 restriction of the observed correlation was observed independently for colorectal and endometrial cancers, although mutation landscapes differed significantly between the two tumor types (Figure 1 B and C).

Last, the absence of a significant correlation among *B2M*-mutant tumors supports the hypothesis that the correlation observed in *B2M*-wild type tumors is related to HLA class I-mediated antigen presentation. The trend observed in *B2M*-mutant tumors may reflect selection events prior to the breakdown of HLA class I machinery; however, we agree that further studies are required to substantiate this mechanistic hypothesis.

As neither GELS nor predicted HLA-A*02:01 epitopes were factoring in our selection of candidates, nor are they likely related to any factor influencing mutation frequency (see detailed discussion below), coincidence or confounding factors as reasons for the four observations mentioned above appear unlikely.

We have expanded the discussion to clarify our interpretation. In addition, we have added new calculations and sections dealing with alternative explanations as discussed separately for the individual topics below.

- The association with other immune evasion mechanisms might provide some additional evidence, but the authors only looked at B2M mutations, and even there, a trend can still be observed. B2M mutations are also known to be quite rare so the question is whether this association can be made for this small subgroup anyway (I couldn't find the B2M mutation frequency in the manuscript BTW, can the authors provide this?). In this regard, I feel the authors should consider looking at associations with other, more common, immune evasion mechanisms such as HLA LOH and PD1/PDL1 amplifications. This would strengthen their claim about immunoediting.

Authors' reply:

We are grateful for the reviewer's comment. In MSI colorectal cancers, *B2M* mutations have been described as the most common immune evasion mechanism, affecting 25 to 30% of tumors. Data from public TCGA and DFCI databases indicate that *B2M* mutations in MSI colorectal cancers occur significantly more frequently than other alterations affecting the HLA class I antigen processing and presentation machinery (Ozcan et al. *Oncolmunology* 2017, PMID 29900056).

In the present study, *B2M* mutation status could be determined in 132 MSI colorectal cancers, and mutations were found in 33 (25%) of the tumors. We thank the reviewer for thorough reading and mentioning that the numbers were lacking in the main text. In the revised version, we have added these numbers in the results section of the manuscript, page 8. In addition, full details about *B2M* mutation status are available in Supplementary Table S2.

Although the correlation with *B2M* mutation status, due to the high frequency of *B2M* mutations in MSI cancer, is expected to be most informative with regard to potential immunoediting, we fully agree with the reviewer that future studies should examine additional immune evasion mechanisms, including HLA LOH and other alterations that require additional techniques for comprehensive analysis.

The results of the present study in our view strongly encourage such efforts, and we are currently expanding the collection of tumor material to pursue this research question.

We have added a sentence to the Discussion, page 11:

“This encourages further studies that systematically address the effects of other immune evasion mechanisms such as alterations of HLA class I heavy chains or components of the antigen processing machinery on antigen landscapes in MSI cancer.”

- *Detecting negative selection boils down to detecting fewer mutations than expected, so an accurate expectation model is always crucial (e.g. metrics like dN/dS all have an expected number in the denominator). As far as I can judge, this is absent in the manuscript. In my understanding, the only underlying expectation is that there should not be a correlation as described in fig. 4C? In their response to reviewer 2 the authors indicate that the reason for not using trinucleotide repeats (or in frame indels) is that they do not lead to frameshifts. Would this be a perfect negative control for the immunoediting claim?*

Authors' reply:

We fully agree with the reviewer that additional analyses help to support the hypothesis of immunoediting. Therefore, we have re-analyzed our data, accounting for the baseline mutation frequency of a given microsatellite.

Many previous studies have demonstrated that the expected mutation frequency for a given microsatellite in MMR-deficient cancers is determined by its length (www.seltarbase.org). Therefore, deviation of the observed mutation frequency of a given microsatellite from the length-specific average has been used as an indicator of positive selection to identify relevant tumor suppressor candidate genes in MSI cancer (Woerner et al. 2003, doi: 10.1038/sj.onc.1206421; Jonchere et al. 2018, doi: 10.1016/j.jcmgh.2018.06.002).

To eliminate microsatellite length as a potential confounding factor, we have calculated for each microsatellite the difference between observed mutation frequency and expected mutation frequency, i.e. the average mutation frequency of all microsatellites of the same length (termed *relative mutation frequency*). Using this approach, we re-analyzed the correlation between GELS and relative mutation frequency, confirming the statistically significant negative correlation in *B2M*-wild type tumors (new Figure S8 A and B). Similarly, HLA-A*02:01 status-related analyses confirmed the previous observations, now even more clearly demonstrating the lack of a significant correlation between predicted HLA-A*02:01 epitope likelihood and mutation frequency in tumors from HLA-A*02:01-negative patients (Fig. S8 C).

We have added a respective paragraph discussing these new analyses in the section “Immunoselection during MSI carcinogenesis” on page 9:

“To account for confounding factors potentially influencing this observation, we investigated a potential relationship between the cMS length and our observed negative correlation. cMS length is a well-known factor influencing the likelihood of indel mutations on the observed mutation frequency 23,27,39 (Fig. S5). However, our analysis demonstrated that GELS was not related to the length of the corresponding microsatellite. Moreover, we replicated the correlation analysis with length-adjusted relative mutation frequencies (relative p_{mut} , computed by subtracting the length-specific M1 average mutation frequency from the observed M1 mutation frequency for each microsatellite), the negative correlation between GELS and relative mutation frequency in *B2M*-wild type tumors was retained (Fig. S8 A and B).”

Due to the substantially different mechanisms contributing to repairing mismatches at mononucleotide and trinucleotide repeats, we do not consider trinucleotide repeats as appropriate negative controls. Whereas the human DNA mismatch repair system is highly effective in correcting mismatches at mononucleotide repeats, its role in trinucleotide repeat alterations is complex. Apparently, mismatch repair can fix short errors occurring during the replication of trinucleotide repeat regions; however, the

correction process seems to be error-prone and can itself introduce long changes in trinucleotide repeat length (Iver et al. Annu Rev Biochem 2016, PMID: 25580529).

- As indicated higher it has been shown that HLA affinities are not independent from mutational signatures (i.e. certain mutations influence specific codons, associated to higher/lower HLA affinities). Wouldn't it be possible that specific indel (ID) signatures result in higher mutation loads in genomic regions encoding non-binding peptides? Showing that this is not the case would again sustain the authors' immunoediting interpretation.

Authors' reply:

The reviewer addresses an important point. We have examined a potential influence of specific indel signatures and did not find evidence for a potential bias introduced by mutational signatures on the observed correlations between immune score and mutation frequency, as discussed in the following. Most importantly, in contrast to point mutation-induced neopeptides, indel-induced neopeptides are not directly related to the change on the nucleotide level, but the likelihood of MHC binding peptides resulting from translational frameshifts is determined by the 3' nucleotide sequence downstream, often distant, from the indel mutation.

For COSMIC Indel signatures (<https://cancer.sanger.ac.uk/cosmic/signatures/ID>), different types of indel mutations at homopolymer stretches (only those were addressed in our study) are distinguished based on three categories, (1) deletion or insertion, (2) how many basepairs affected by deletion or insertion (1 bp vs > 1 bp), (3) length of the homopolymer stretch, i.e. in our case length of the coding microsatellite. As all microsatellites examined in our study consist of more than 5 repeat units, and observed mutations were mostly deletions of single base pairs, the only potential influence of indel signatures may be related to differences between G/C and T/A microsatellite stretches. We have analyzed the GELS of neopeptides resulting from indels affecting G/C and T/A repeats and did not detect any significant difference (median GELS for A/T repeats: 0.617, median GELS of G/C repeats: 0.615).

These data and a brief discussion about a potential influence of indel signatures have been added to the manuscript, pages 11 and 12.

I understand that new approaches might be hard to implement at this stage of the manuscript, but I feel these concerns should minimally be addressed in the discussion section. I would also be more prudent with the claim that the findings are related to immunoediting (especially in the title) without further evidence.

Authors' reply:

We are very grateful and tried to thoroughly revise our manuscript, mentioning alternative explanations for our findings (Main text, page 9, Discussion, pages 11 and 12). In addition, we have toned down the title statement, now reading "The shared frameshift mutation landscape of MSI cancers suggests immunoediting during tumor evolution". Moreover, specific sections of the manuscript have been adjusted accordingly, for example Discussion, page 12: "

"... we for the first time are able to provide evidence that the cMS mutation patterns in MSI cancer show signs of immune selection ..."

now reading

“... we for the first time identified patterns in the cMS mutation spectrum of MSI cancers suggestive of immune selection ...”

2) While some validation was added on cell lines, I still miss some benchmarking for the ReFrame tool. How does the tool perform compared to NGS with state-of-the-art indel callings? Is the high indel frequency realistic? Is there a possibility that FFPE somehow influenced this frequency? (As the authors indicate in their responses to all reviewers, HLA typing was not possible on FFPE tissues, so I assume there were DNA quality problems?)

Authors' reply:

We thank the reviewer for this question. Multiple previous publications, including the most comprehensive data collection from the literature (Woerner et al. *Nucleic Acids Res* 2010) and independent other studies (Alhopuro et al. *Int J Cancer* 2012; Jonchere et al. 2018, doi: 10.1016/j.jcmgh.2018.06.002) reported mutation frequencies similar to our study, supporting that the detected mutations are not artificial.

As in most previous studies on somatic microsatellite mutations in MMR-deficient cancers, we used fragment length analysis, a long-established tool for MSI detection in scientific studies and molecular diagnostics. Fragment length analysis was used as a gold standard for indel detection at microsatellites by a recent study (Jonchere et al. 2018, doi: 10.1016/j.jcmgh.2018.06.002), in which direct comparison with NGS using state-of-the-art indel calling demonstrated high concordance. Thus, the authors concluded that the mutations observed in their study were not artifacts introduced by the one or the other method.

In our study, adding ReFrame as a novel analysis tool allowed (1) quantification of mutant alleles, and thereby (2) differentiation between M1 frame and M2 frame, which was essential to specifically examine the immunological consequences.

It is unlikely that formalin fixation had a significant influence on the results, because of the following reasons:

- Cell lines established from MSI CRC specimens show even higher indel mutation frequencies than archival FFPE CRC samples using the same mutation detection tools (www.seltarbase.org);
- typical sequencing artifacts related to formalin fixation reported in the literature are rather base exchanges (e.g. C>T) occurring at a low variant allele frequency (Do and Dobrovic *Clinical Chemistry* 2015, doi:10.1373/clinchem.2014.223040);
- to minimize a potential influence of formalin fixation on the results, we ensured that buffered formalin was used during tissue fixation (avoiding the generation of abasic sites), fixation times as far as known were below 48 hours;
- DNA was isolated by microdissection from whole tumor areas for every available section, thus large amounts of template DNA could be obtained, which greatly reduces the risk of false mutation calling due to formalin fixation.

These points have been added to the discussion, page 12.

3) The authors claim that their findings are HLA type-dependent, so I expected an HLA genotyping for all patients. I understand from the responses to all 3 other reviewers that this was not technically feasible. I think this should also be added to study limitations in the discussion section and more importantly, any claims that the findings are HLA type-specific should be avoided as they are misleading. Showing a correlation for the most common HLA0201 cannot be generalized in this way. In this regard, I wonder

whether the fact that HLA0201 is one of the strongest peptide binding alleles might have had an influence on the results?

Authors' reply:

We agree with the reviewer that the findings related to the presence or absence of HLA-A*02:01 cannot be generalized to all HLA types. We cannot exclude the possibility that different HLA binding properties differentially affect immunoediting for different HLA alleles.

We have revised the manuscript according to the reviewer's suggestion, only referring to HLA-A*02:01, for example Discussion, page 13:

"The observation of HLA type-dependent immunoediting during the development of MMR-deficient cancers also implies ..."

now reading

"The observation of HLA-A*02:01-dependent immunoediting during the development of MMR-deficient cancers also implies ..."

In addition, the limitation regarding the restriction to HLA-A*02:01 is mentioned on page 13:

"Because only paraffin-embedded tissue was available from CRC and EC samples, HLA typing could only detect absence or presence of HLA-A*02:01, restricting HLA-related immunoediting analyses to HLA-A*02:01."

Some other, minor comments:

- NMD (nonsense-mediated mRNA decay) is expected to act on indels and I was surprised that this was not addressed in the manuscript. It might actually offer an explanation on why some TSGs behave as outliers (e.g. no expression implies no immune selection).

Authors' reply:

The reviewer's point is well taken. NMD is known to differentially affect the degradation of target mRNAs in MSI tumors thereby generating NMD-resistant as well as NMD-sensitive frameshift-mutated target mRNAs (El-Bchiri et al. 2008, 10.1371/journal.pone.0002583; Williams et al. 2010, doi: 10.1371/journal.pone.0016012; Bokhari et al. 2018, doi: 10.1038/s41389-018-0079-x). Accordingly, frameshift-mutant mRNAs that escape NMD surveillance are expected to account for the high frameshift peptide load in MSI tumor cells. On the other hand, NMD elimination of mutant transcripts can interfere with their expression and recognition of the respective mutant peptides by the immune system.

However, complete elimination of mutant mRNAs by NMD is rare, as for several of the candidates identified as outliers expression of mutant mRNAs was demonstrated previously by our group and others. Notably, even NMD sensitivity does not necessarily imply lack of frameshift peptide presentation to the immune system. Since NMD requires a pioneer round of protein translation, minor amounts of frameshift peptides can still be generated, and *in vitro* evidence suggests that a single peptide-MHC complex on a target cell can elicit a cytolytic T-cell response (Sykulev et al. 1996, doi: 10.1016/s1074-7613(00)80483-5). In fact, peptides generated during this pioneer round of protein translation have been identified as a relevant source of antigenic peptides presented via the MHC class I complex (Apcher et al. 2011, doi: 10.1073/pnas.1104104108). We fully agree with the reviewer that systematic studies addressing the impact of NMD on systemic frameshift peptide-specific immune responses in MMR-deficient cancer are of high importance.

We have added a respective paragraph referring to NMD to the discussion, page 12. In addition, we have added a statement to the Limitations section, page 13:

“... and a possible influence of NMD could not be systematically addressed.”

- *Immunoediting or immuno-editing? The authors use both ways of writing.*

Authors' reply:

We have removed the hyphen, now using the term immunoediting consistently throughout the manuscript.

- *Why is the neo in neoantigens italicized in the manuscript?*

Authors' reply:

We have changed the formatting to neoantigens, not using italics.

- *Gene names should be italicised.*

Authors' reply:

We are grateful for the comment and have adapted the formatting.

- *Beginning of last paragraph p. 7. “... to distinguish mutation types ...”. I was initially confused, thinking about mutation variants (e.g. missense mutations). It gets clearer later but talking about indel mutation types might avoid some confusion for the reader.*

Authors' reply:

We agree and have changed the sentence and the Legend to Table 1 accordingly.

- *Figure 3. It was not immediately clear to me what the difference between the left/middle/right figures in each panel was. I would add some label to clarify, or even showing only Kd 50nM given the redundancy between the figures.*

Authors' reply:

We have added labels for clarification as suggested. Similarly, we have adapted Supplementary Figure 3. As there are several recent studies that suggest relevance of lower affinity MHC binding peptides, we would prefer retaining the illustration for different cutoff values (see also below).

- *An HLA affinity Kd cut-off of 50 nM or 500nM is commonly used. I haven't seen 5000nM before and don't think this is too useful.*

Authors' reply:

A recent study by Bonsack et al. (Cancer Immunology Research 2019) evaluated experimentally MHC binding of predicted peptides and demonstrated that using a threshold of < 500 nM misses a significant proportion of actual MHC binding peptides, depending on the MHC molecule more than 50% (Figure 4). Future studies need to examine the relevance of such peptides. We prefer keeping the figure in its

current layout for completeness, including also < 5000 nM predictions. We have added labels to the respective parts as discussed above.

- When mentioning in the discussion that other studies (refs 60,61) failed to detect negative selection and immunoediting, the authors' interpretation is superior resolution of ReFrame. I would like to note that most previous studies only focused on single nucleotide substitutions, so I don't agree completely on the interpretation.

Authors' reply:

We are grateful to the reviewer and have changed the respective section, now accounting for this important difference.

- Several choices became clear to me after reading the rebuttal but not the manuscript. I think the authors should try to clarify these in the manuscript as well (e.g. they respond to reviewer 2 that 40 genes is a limitation of the study, but this is not mentioned when discussing study limitations in discussion; same for technical difficulties with HLA typing as I mentioned higher)

Authors' reply:

We have incorporated statements in the Discussion section, explaining topics like candidate selection according to our previous conversation with reviewer 2 (page 12).

Additional changes:

We have added Damian Stichel who has worked on data re-analysis and code generation as a coauthor.

REVIEWERS' COMMENTS:

Reviewer #5 (Remarks to the Author):

The authors have adequately addressed my comments. Although I'm still not 100% convinced about the immunoediting interpretation, I'm pleased to see additional clarifications and critical notes in the manuscript.

I wish to congratulate the authors with this interesting study and look forward to reading their future work on the subject.